# A fast approximation for 1D Inversion of Transient Electromagnetic Data by BP Neural Network and improved Particle Swarm Optimization

Ruiyou Li, Huaiqing Zhang[*], Nian Yu, Ruiheng Li, Qiong Zhuang

The State Key Laboratory of Transmission Equipment and System Safety and Electrical New Technology, Chongqing University, Chongqing, 400044, China

[*]*Correspondence to*: zhanghuaiqing@cqu.edu.cn

**Abstract.** As one of the most active nonlinear inversion methods in transient electromagnetic (TEM) inversion, the back propagation (BP) neural network has high efficiency because the complicated forward model calculation is unnecessary in iteration. The global optimization ability of the particle swarm optimization (PSO) is adopted for amending BP's sensitivity on initial parameters, which avoids it falling into local optimum. A chaotic oscillation inertia weight PSO (COPSO) is proposed in accelerating convergence. The COPSO-BP algorithm performance is validated by two typical testing functions, then by two geoelectric models inversion and a field example. The results show that the COPSO-BP method has better accuracy, stability and relative less training times. The proposed algorithm has a higher fitting degree for the data inversion, and it is feasible in geophysical inverse applications.

**Keywords**:transient electromagnetic inversion; BP neural network; particle swarm optimization; chaotic oscillation

## 1 Introduction

Transient electromagnetic (TEM) method applies the secondary receiving voltage induced by the rapid switching off pulse current, and then deduces the geoelectrical parameters consisting of the resistivities and thicknesses of the layers. The later is a typical TEM inversion issues with nonlinear feature. The linear inversion method was simple and widely used through linearization process, yet it is extremely dependent on initial parameters selection and resulting in poor inversion accuracy. Hence, the nonlinear inversion methods attract more geophysicists attention in recent years.

The artificial neural network(ANN) is one of the most active nonlinear inversion methods, it has

---

**Conflicts of Interests**

The authors declare that they have no conflict of interest.

very high computation efficiency because the complicated forward model calculation is unnecessary in iteration. All the geoelectrical parameters and the forward model relations are implied in the weight and threshold parameters of ANN. And it is different from the non-linear Monte Carlo method with global space search solution (He et al., 2018; Jha et al., 2008; Pekşen et al., 2014; Sharma, 2012; Tran and Hiltunen, 2012). Srinivas et al. (2012) compared the inversion performance of BP, radial basis function(RBF) and generalized regression neural network (GRNN) in vertical electrical sounding data, then established a 1-D inversion model with BP and finally realized the parameters inversion. Maiti et al. (2012) proposed a Bayesian neural network training method in 1-D electrical sounding. Jiang et al. (2018) improved the training method for kernel principal component wavelet neural network and achieved the resistivity imaging. Jiang et al. (2016a) gave a learning algorithm based on information criterion (IC) and particle swarm optimization for RBF network which improves the global search ability. Johnson (2017) utilized neural network method to invert multi-layer georesistivity sounding. Jiang et al. (2016b) presented a pruning Bayesian neural network (PBNN) method for resistivity imaging and solved the instability, local minimization problems. Raj et al. (2014) solved non-linear apparent resistivity inversion problems with ANN. The ANN has been widely applied in electric prospecting data interpretation for its powerful fitting ability. However, the neural network method is sensitive to initial parameter settings and falls easily into local minimum. Lots improved methods were proposed for balancing the convergence rate and inversion quality. Zhang and Liu (2011) proposed ant colony optimization for ANN and applied in high density resistivity, acquired smaller inversion errors and higher determinant coefficients. Dai et al. (2014) suggested a differential evolution (DE) for BP which enhanced the global search ability. Marina et al. (2014) introduced the genetic algorithm for ANN.

The Particle swarm optimization (PSO) has simple structure, fast convergence rate, high accuracy and global optimization ability. Fernández et al. (2010) successfully introduced the PSO in 1-D resistivity inversion. Godio and Santilano (2018) applied it in geophysical inversion and deduced a depth resistivity earth model. Since the PSO's global searching performance, the BP's initial weights and thresholds can be trained by PSO and then the BP's global optimization ability can be improved. Comparing to the standard PSO (SPSO), a chaotic oscillation inertia weight PSO (COPSO) which can accelerate the convergence rate in the early stage was proposed naturally(Shi et al., 2009).

The paper structure is as following: the principle of PSO algorithm with different inertia weights schemes, the BP neural network and the proposed COPSO-BP algorithm are given in section 2. Then, the COPSO-BP algorithm performance is validated by two typical testing functions in section 3. And in later section, inversion simulations of a three-layer and five-layer geoelectric models are carried out, the hidden layer neuron numbers determining method is put forward and algorithms performance is compared.

## 2   Principle of COPSO-BP Algorithms

### 2.1   Chaotic Oscillation PSO algorithm

For $N$-dimensional optimization problem, supposing the position (resistivity and thickness for layered model parameters inversion) and velocity(update speed) of the $i$-th particle (global search group number) at time $t$ are $x_i = (x_{i1}, x_{i2}, \cdots, x_{iN})$ and $v_i = (v_{i1}, v_{i2}, \cdots, v_{iN})$ respectively. Then, at time $t+1$ ,they can be calculated by the iterations as

$$v_{id}^{t+1} = \omega \cdot v_{id}^t + c_1 r_1 (p_{id}^t - x_{id}^t) + c_2 r_2 (p_{gd}^t - x_{id}^t) \tag{1}$$

$$x_{id}^{t+1} = x_{id}^t + v_{id}^{t+1} \tag{2}$$

where $r_1, r_2$ are random value evenly distributed in the interval $(0,1)$, $c_1, c_2$ are learning factors (usually equal to 2). And $p_{id}$, $p_{gd}$ means the individual and global maximum.

The inertia weight parameter $\omega$ affects the algorithm performance seriously. A fixed weight always was used in the early time, and then various dynamic weights were proposed. Shi et al. (2010) have summarized several methods as

$$\omega_1(t) = \omega_s - (\omega_s - \omega_e) t / T_{\max} \tag{3}$$

$$\omega_2(t) = \omega_s - (\omega_s - \omega_e)(t / T_{\max})^2 \tag{4}$$

$$\omega_3(t) = \omega_s - (\omega_s - \omega_e)\left[ 2t/T_{\max} - (t/T_{\max})^2 \right] \tag{5}$$

Where $\omega_s$ and $\omega_e$ are the start and end weight. The $t$, $T_{\max}$ are the current and maximum iteration. The above weights are of smooth and monotonically decreasing. In this paper, we proposed a decreasing oscillation weights scheme which was based on chaotic logistic equation. Its specific calculation formula as

$$x_{t+1} = \mu x_t (1 - x_t) \qquad t = 0, 1, 2, \cdots, n \tag{6}$$

$$\omega_c(t) = \omega_e + (\omega_s - \omega_e)(0.99^t \cdot x_t) \tag{7}$$

where $\mu$ is the control parameter. A complete chaos state is established for $x \in (0,1)$ and $\mu = 4$, an inertia weight is then obtained from Eq.(7). Numerical experiments were carried out correspondingly and showed that the initial value of $x_0$ has little effect on inertia weight $\omega$. The inertia weights comparison was shown in Fig.1 where $x_0 = 0.234$ and $\mu = 4$ for chaotic oscillation.

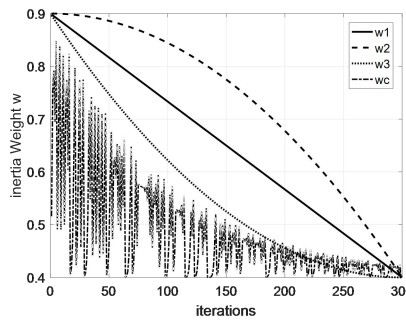

**Fig. 1** Inertial weight curves comparison
**2.2   BP Neural Network**
BP neural network is multi-layer feed forward structure, and a typical three-layer network is
shown in Fig. 2 (Yong et al., 2009).

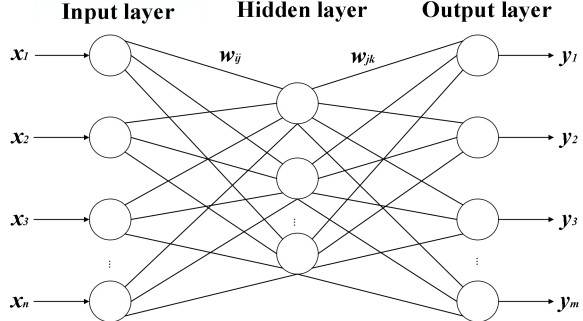

**Fig. 2** Three-layer BP neural network structure
where $x_1, x_2, ..., x_n$ are the input value, $y_1, y_2, ..., y_m$ are the predicted output, $w_{ij}, w_{jk}$ are the network
weights. The threshold parameter $\alpha$ is defined in hidden layer with its output
$$H_j = f\left(\sum_{i=1}^{n} w_{ij} x_i - \alpha_j\right) \qquad j = 1, 2, \cdots, l \tag{8}$$

where $l$ is the hidden layer nodes numbers, $f$ is the activation function with different expressions,
and the most widely used is sigmoid type function. The predicted output for the $k$-th unit is
calculated by
$$O_k = \sum_{j=1}^{l} H_j w_{jk} - b_k \tag{9}$$

And parameter $b$ means the output threshold. Then the prediction error can be determined based
on predicted output $O_k$ and the expected output $T_k$ as $e_k = (T_k - O_k)O_k(1 - O_k)$. The updating formula
for weights and thresholds are as following
$$\begin{cases} w_{ij} = w_{ij} + \eta H_j(1 - H_j) x_i \sum_{k=1}^{m} w_{jk} e_k \\ \qquad w_{jk} = w_{jk} + \eta H_j e_k \\ \alpha_j = \alpha_j + \eta H_j(1 - H_j) \sum_{k=1}^{m} w_{jk} e_k \\ \qquad b_k = b_k + e_k \end{cases} \tag{10}$$

where $i = 1, 2, ..., n$; $j = 1, 2, ..., l$; $k = 1, 2, ..., m$; and $\eta$ is the learning rate.
**2.3   BP Neural Network with COPSO algorithm**
The initial parameters are chosen randomly, which affects the convergence rate, learning
efficiency and perhaps falling into local minimum. The Chaotic Oscillation PSO (COPSO) has a
much better global optimization capability, therefore, the COPSO algorithm is proposed to optimize
the initial weight and threshold of BP. The COPSO-BP pseudo-codes were briefly described as
following:

**Table.1** Pseudo-codes of COPSO-BP algorithm

| | |
|---|---|
| 1: | *BP network structure definition* (neuron numbers *n,l,m,* and *activation function*) |
| 2: | *COPSO initialization for BP* (weights, threshold as *X*. PSO parameters as $V_{min},V_{max},\omega_c,c_1,c_2$, size *M*, $T_{max}$) |
| 3: | *Initializing BP* with $X_i$ (*i=1,2,...,M*) and *evaluating fitness* by Eq.(11) for each individual |
| 4: | Setting the $p_{id}$ and $p_{gd}$ |
| 5: | **While** *iter*< $T_{max}$ **do** |
| 6: | updating inertia weight by Eq.(7) |
| 7: | **for** *i=1:M* (all particles) **do** |
| 8: | updating velocity $V_i$ by Eq.(1) |
| 9: | updating particle position $X_i$ by Eq.(2) |
| 10: | *Initializing BP* with new $X_i$ and *calculating fitness* by Eq.(11) |
| 11: | **if** $X_i$ is better than $p_{id}$ |
| 12: | Set $X_i$ is to be $p_{id}$ |
| 13: | **End if** |
| 14: | **if** $X_i$ is better than $p_{gd}$ |
| 15: | Set $X_i$ is to be $p_{gd}$ |
| 16: | **End if** |
| 17: | **End for *i*** |
| 18: | *iter = iter*+1 |
| 19: | **End While** |
| 20: | *Initializing BP* with $p_{gd}$ |
| 21: | *Inputting and obtaining the predicted output* |

The formula for calculating the *i*-th particle fitness is defined as
$$f_i = \frac{1}{S}\sum_{s=1}^{S}\sum_{j=1}^{m}\left(Y_{sj} - \hat{Y}_{sj}\right)^2 \tag{11}$$
where *S* is the number of training set samples, *m* is the output neurons number, $Y_{sj}$ is the *j*-th true
output of the *s*-th sample, and $\hat{Y}_{sj}$ is the corresponding predict output.
## 3   Algorithm Testing
In order to investigate the COPSO-BP performance and reliability, Rosenbrock and Bohachevsky
testing functions were adopted, which are typical non-convex functions and mainly to evaluate the
performance of unconstrained algorithms. However, due to the random nature of the function, it is not
easy to solve and has a global minimum function value of zero.
(1) *Rosenbrock* function:

$$f_1(x) = 100 \times \left(x_1^2 - x_2\right)^2 + \left(1 - x_1\right)^2, x_i \in \left[-10,10\right], i = 1,2 \tag{12}$$

(2) *Bohachevsky* function:

$$f_2(x) = x_1^2 + x_2^3 - x_1 x_2 x_3 + x_3 - \sin\left(x_2^2\right) - \cos\left(x_1 x_3^2\right), x_i \in \left[-2\pi, 2\pi\right], i = 1,2,3 \tag{13}$$

The standard PSO-BP (SPSO-BP) with linear decreasing inertia weight as Eq.(3), the
COPSO-BP were carried out respectively. The three-layer BP of *n-s*-1 structure is constructed with
different hidden nodes. The PSO parameters are population size $M = 60$, learning factors $c_1 = c_2 =$
2.0, the maximum iteration $T_{max} = 30$, inertia weight $\omega_s = 0.9$, $\omega_e = 0.4$, $x_0 = 0.234$ and $\mu = 4$ for
chaotic parameters, the search dimension $D = n \times s + s \times 1 + s + 1$ which includes all the neuron
weights and thresholds. For BP network, 150 training samples and 50 testing samples were
randomly produced within the variable range. The training error is defined as

$$E = \frac{1}{S} \sum_s^S \left(T_s - O_s\right)^2 \tag{14}$$

where $S$ is the training samples number, $T_s$, $O_s$ are the expected and predicted output for training
sample $s$ respectively. The network structures with minimum training errors for *Rosenbrock* and
*Bohachevsky* functions are 2-7-1 and 3-6-1 respectively. The simulation performs 20 times for
each testing function with SPSO-BP and COPSO-BP algorithms. The numerical result was shown
in Table.2. One of the evolutionary training error curves (select one in 20 times randomly) were
shown in Fig.3, and the fitting curves of COPSO-BP algorithm were shown in Fig.4.
**Table.2** Comparison of SPSO-BP and COPSO-BP algorithm for testing functions

| Testing functions | SPSO-BP | | COPSO-BP | |
|---|---|---|---|---|
| | Average value | Optimal value | Average value | Optimal value |
| *Rosenbrock* | 2.375e-3 | 2.300e-5 | 1.201e-3 | 2.410e-06 |
| *Bohachevsky* | 0.225 | 1.024e-3 | 0.193 | 3.360e-4 |

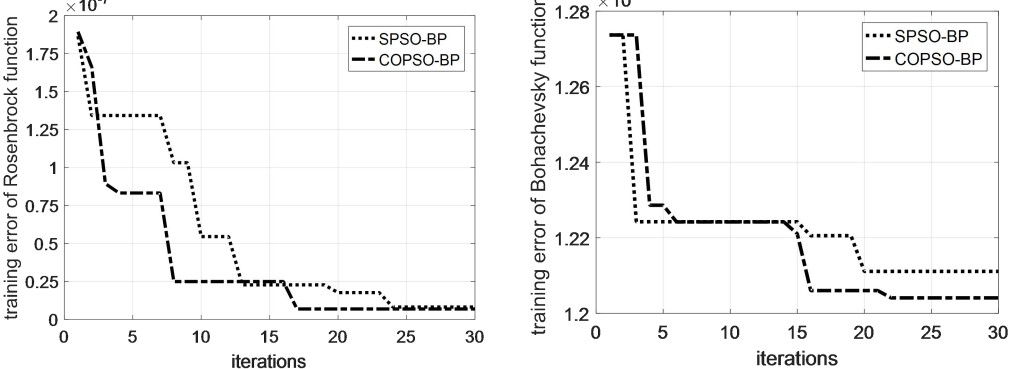


**Fig. 3** Training error curves of SPSO-BP and COPSO-BP algorithms

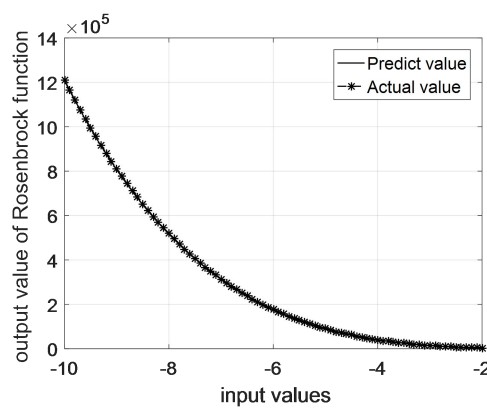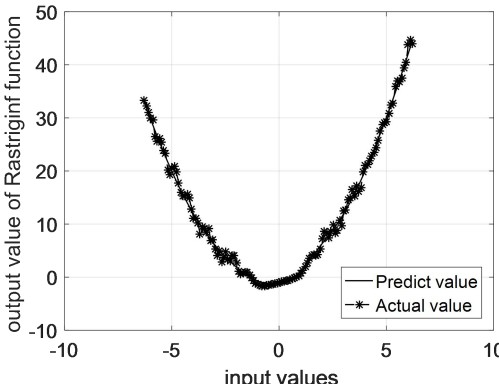

**Fig. 4** Fitting curves of COPSO-BP algorithm

It can be seen in Table.2 that both the SPSO-BP and COPSO-BP algorithms can acquire a relative high accuracy for testing functions, the COPSO-BP is a slightly better than SPSO-BP. However, the COPSO-BP has better convergence rate and optimization efficiency in the early stage in Fig.3. Therefore, the SPSO-BP and COPSO-BP algorithms have strong learning ability, good stability and generalization ability, which will be suitable for TEM inversion.

## 4 Layered model and parameter analysis

### 4.1 Forward Model

According to Kaufman's derivation (1983), the frequency response of central loop source for the layered model takes the following Hankel transform

$$H_z(\rho,\omega) = Ia \int_0^\infty \frac{m^2}{m + m_1/R_1^*} J_1(m\rho)\,\mathrm{d}m \tag{15}$$

where $a$ is the radius of transmitting coil, $I$ is the excitation current, $\rho$ is the center distance between the transmitting coil and the receiving coil, $J_1(m\rho)$ is the first-order Bessel function, $m$ is integral variable, $m_1 = (m^2-k_1^2)^{1/2}$, $k_1$ is the conduction current, $\sigma_1$ is the conductivity, $k_1 = -i\omega\mu\sigma_1$, and $R_1^*$ is the first layer apparent resistivity conversion function which can be obtained by the following recurrence formula

$$\begin{cases} R_n^* = 1 \\ R_j^* = \dfrac{m_j R_{j+1}^* + m_{j+1}\mathrm{th}\left(m_j h_j\right)}{m_{j+1} + m_j R_{j+1}^*\mathrm{th}\left(m_j h_j\right)} \end{cases} \tag{16}$$

There is no analytical solution for the time-domain response for layered model, it can only be solved by numerical calculation. The Hankel transform in formula (15) is calculated by an improved digital filtering algorithm with 47 points $J_1$ filter coefficient, and then time response can be obtained using the Gaver-Stehfest transform as follows:

$$H_z(\rho,t) = \frac{\ln 2}{t} \sum_{n=1}^{N} K_n H_z(\rho,s_n) \tag{17}$$

where $s_n = (ln2/t) \times n$, $K_n$ is the coefficient, N is determined by the computer bits, generally N=12.

The ramp excitation current of TEM is

$$I(t) = \begin{cases} 0, & t<0 \\ t/T_1, & 0 \le t<T_1 \\ 1, & T_1<t \end{cases} \tag{18}$$

where $T_1$ is the turn-off time, and the Laplace transform is

$$I(s) = \frac{1}{T_1 s^2} - \frac{1}{T_1 s^2} e^{-T_1 s} = \frac{1}{T_1 s^2}(1 - e^{-T_1 s}) \tag{19}$$

Therefore, for a specific layered model, the apparent resistivity conversion function $R_1^*$ is firstly calculated by recurrence formula (16) based on geoelectric structure parameters. And then the frequency response at fixed point $H_z(\omega)$ is calculated by Hankel transform as formula (15). For ramp excitation, the Laplace transform of $H_z(s)$ should multiplied by $I(s)$. Finally, the time response $H_z(t)$ is obtained by Gaver-Stehfest transform as formula (17). So the $H_z(t)$ is obtained by a Gaver-Stehfest transform, a Hankel transform and a recurrence calculation, and it is somewhat heavy computational consuming.

However, the vertical magnetic field $H_z(t)$ is the actual observed signal in transient electromagnetic method in engineering applications. It is the inversion input and output is geoelectric structure parameters. A method which can avoid the complicated forward model calculation is of great importance in algorithm efficiency.

## 4.2 BP network design and COPSO algorithm

For BP structure, the output nodes are determined by the number of inversion geoelectrical parameters, the input nodes are determined by the samples number of $H_z(t)$, the hidden nodes varies according to approximation performance. As a three-layer or five-layer geoelectric model, its geoelectrical parameters are 5 (three resistivity and two thickness parameters) or 9 (five resistivity and four thickness parameters), the output nodes are 5 or 9 correspondingly. The characteristic samplings of $H_z(t)$ are chosen as 10 or 20, which are determined by the model's complexity, more layers mean mores sampling points needed. The 10 samplings were selected in this paper hence with 10 input nodes. While for the hidden layer neuron, its number is related to the weights and threshold parameters amount directly and affects the BP performance greatly. An appropriate hidden nodes number is necessary and a determination coefficient $R^2$ is defined for evaluating as

$$R^2 = \frac{\left( n\sum_{i=1}^{n} Y_i \hat{Y}_i - \sum_{i=1}^{n} Y_i \sum_{i=1}^{n} \hat{Y}_i \right)^2}{\left( n\sum_{i=1}^{n} \hat{Y}_i^2 - \left( \sum_{i=1}^{n} \hat{Y}_i \right)^2 \right)\left( n\sum_{i=1}^{n} Y_i^2 - \left( \sum_{i=1}^{n} Y_i \right)^2 \right)} \tag{20}$$

where $Y_i$ is the true value, $\hat{Y}_i$ is the predicted value for $i$-th training data, $n$ is the training data number. A larger determination coefficient means a better approximation performance. The

simulations on hidden nodes effect were carried out for a three-layer and five-layer geoelectric models. The BP structure is 10-*s*-5 and 10-*s*-9, its transfer, training and learning functions are 'Log sigmodial', 'Levenberg-Marquardt' and 'Gradient descent momentum' respectively. The average, minimum and maximum value of $R^2$ were obtained after running 20 times for each simulation. The $R^2$ curves were shown in Fig.5.

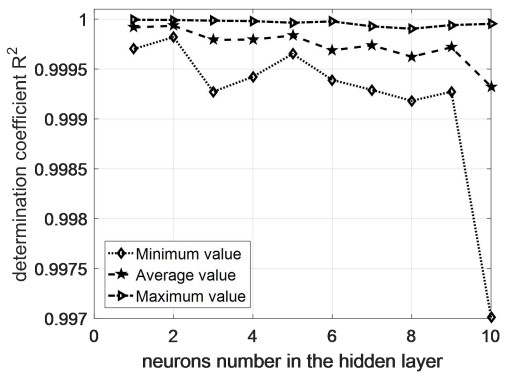 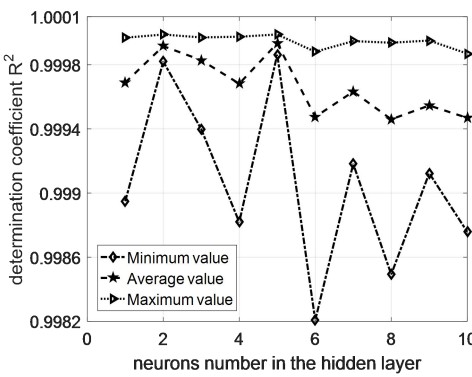

(a)Three-layer geoelectric model            (b)Five-layer geoelectric model

**Fig. 5** Influence of hidden layer nodes on $R^2$ for different geoelectric model

It can be seen that the optimal neural network structures were 10-2-5 and 10-5-9 for three and five-layer models based on the maximum $R^2$. Then, the PSO-BP algorithms with different inertia weight were implemented and compared for three-layer model. The BP structure was chosen as 10-2-5, four types of inertia weight as Eq. (3~7) in PSO were compared in Table.3.

**Table.3** Comparison of different inertia weights in PSO algorithms ( $\omega_s = 0.9$, $\omega_e = 0.4$ )

| inertia weight | iteration number | minimum fitness | average fitness | convergence time(s) |
|---|---|---|---|---|
| $\omega_1$ | 9 | 1.3914e-3 | 1.3982e-3 | 65.21 |
| $\omega_2$ | 29 | 1.4406e-3 | 1.4418e-3 | 204.97 |
| $\omega_3$ | 25 | 1.4168e-3 | 1.4224e-3 | 189.17 |
| $\omega_c$ | 6 | 1.3846e-3 | 1.3925e-3 | 44.34 |

The simulation was implemented on Core (TM) i5-7500 with 8GB memory. It is obviously found in Table.3 that the COPSO algorithm has much faster convergence rate, less iteration number and time consuming.

**4.3  Layered model inversion**

A 3-layered and 5-layered geoelectric models were investigated, which the PSO parameter values are the same as those of the Algorithm Testing parts in the paper. In order to simulate actual TEM applications, the ramp turn-off is taken into account. Considering the probability distribution characteristic of above algorithms, the average of 20 simulation results is chosen. The BP, SPSO-BP, COPSO-BP algorithms and non-linear programming genetic algorithm (NPGA) (Li et al., 2017) were compared.

**(1) *3-layered H type model***

The central loop TEM parameters are set as following, transmitting coil radius $a$ = 100 m, ramp
emission current is 100 A, turn-off time is 1 μs. In the geoelectric model, the resistivity $\rho_1$ = 100
Ω·m, $\rho_2$ = 10 Ω·m, $\rho_3$ = 100 Ω·m and thickness $h_1$ = 100 m, $h_2$ = 200 m.
The BP training samples which is a series of $H_z(t)$ for different geoelectrical parameters were
generated by TEM forward model. The resistivity ranges were $\rho_1 \in (50,150)$, $\rho_2 \in (5,15)$,
$\rho_3 \in (50,150)$, the thickness range were $h_1 \in (50,150)$, $h_2 \in (100,300)$, and choosing 1000 random
groups. The resistivity and thickness distributions of $\rho_1$ and $h_1$ were shown in Fig.6. The relative
error is defined as
$$Err_{\_rel} = \left| \frac{T^*_{\_cal} - O^*_{\_ref}}{O^*_{\_ref}} \right| \qquad (21)$$
where $T^*_{\_cal}$, $O^*_{\_ref}$ are the calculated and reference value for the geoelectric models.

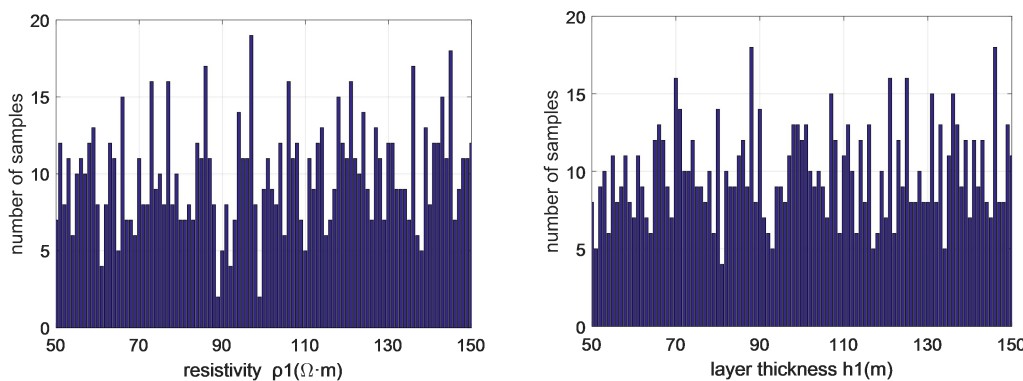


**Fig. 6** Distribution of resistivity $\rho_1$ and thickness $h_1$ in training samples
The inversion results were shown in Table.4. and Fig.7~8. The BP type algorithms were
superior to the NPGA inversion in Table.4. Moreover, the inversion accuracy, convergence rate
and optimization ability of the COPSO-BP algorithm were better than others.
**Table.4** Inversion comparison of three-layer H type geoelectric model

| H type | resistivity $\rho$ (Ω·m) | | | thickness $h$(m) | | total relative error(%) |
|---|---|---|---|---|---|---|
| | $\rho_1$ | $\rho_2$ | $\rho_3$ | $h_1$ | $h_2$ | |
| true values | 100 | 10 | 100 | 100 | 200 | -- |
| BP relative error(%) | -0.275 | -0.625 | 0.765 | -0.968 | -0.649 | 3.284 |
| SPSO-BP relative error(%) | 0.062 | -0.322 | -0.737 | -0.579 | -0.970 | 2.672 |
| COPSO-BP | 100.031 | 9.991 | 99.310 | 100.234 | 200.886 | -- |
| COPSO-BP relative error(%) | 0.031 | -0.087 | -0.689 | 0.234 | 0.443 | 1.487 |
| NPGA relative error(%) | 0.133 | -0.034 | 3.450 | -7.305 | -0.401 | 11.323 |

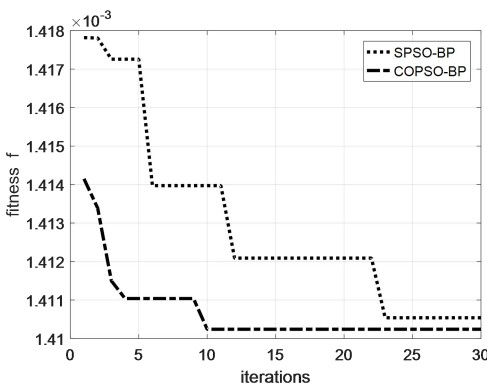 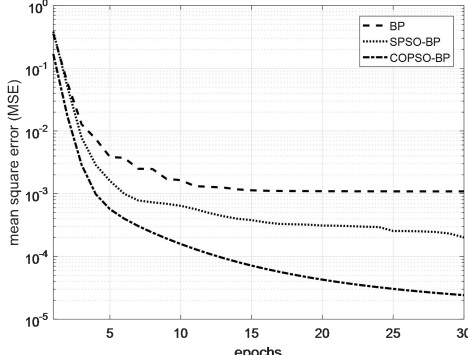


**Fig. 7** Fitness curves of SPSO-BP and COPSO-BP    **Fig. 8** Mean square error curves comparison
Additional results showed that the solution range of $\rho_1$ and $h_1$ in 20 times simulations for above
algorithms were $\rho_1 \in (97.980, 103.102)$, $h_1 \in (96.962, 102.480)$ for BP, $\rho_1 \in (98.954, 101.137)$,
$h_1 \in (96.955, 101.829)$ for SPSO-BP, $\rho_1 \in (99.382, 100.989)$, $h_1 \in (97.877, 101.044)$ for COPSO-BP
respectively. Therefore, the COPSO-BP can acquire higher accuracy and is more stable.
**(2) *5-layered KHK type model***
A 5-layered KHK type geoelectric model was adopted and its resistivity were $\rho_1 = 100 \ \Omega \cdot m$, $\rho_2 =$
$\Omega \cdot m$, $\rho_3 = 50 \ \Omega \cdot m$, $\rho_4 = 200 \ \Omega \cdot m$, $\rho_5 = 30 \ \Omega \cdot m$ and thickness were $h_1 = 100$ m, $h_2 = 200$ m, $h_3$
$= 300$ m, $h_4 = 500$ m.
The training samples with parameter ranges were $\rho_1 \in (50, 150)$, $\rho_2 \in (150, 450)$, $\rho_3 \in (25, 75)$,
$\rho_4 \in (100, 300)$ , $\rho_5 \in (15, 45)$ for resistivity, and $h_1 \in (50, 150)$, $h_2 \in (100, 300)$, $h_3 \in (150, 450)$,
$h_4 \in (250, 750)$ for thickness. The 1000 groups training samples were generated within above
ranges. The inversion results were shown in Table.5 and Fig.9~10. As can be seen that the
COPSO-BP algorithm has better global optimization performance.
**Table.5** Inversion comparison for five-layer KHK type geoelectric model

| KHK type | resistivity $\rho(\Omega \cdot m)$ | | | | | thickness $h$(m) | | | | Total relative error(%) |
|---|---|---|---|---|---|---|---|---|---|---|
| | $\rho_1$ | $\rho_2$ | $\rho_3$ | $\rho_4$ | $\rho_5$ | $h_1$ | $h_2$ | $h_3$ | $h_4$ | |
| true values | 100 | 300 | 50 | 200 | 30 | 100 | 200 | 300 | 500 | -- |
| BP relative error(%) | -1.006 | -0.862 | -1.014 | -0.030 | 1.119 | -0.362 | -0.298 | -0.575 | -0.376 | 5.645 |
| SPSO-BP relative error(%) | 0.429 | 1.040 | -0.577 | -0.071 | -0.883 | -0.002 | 0.657 | -0.655 | -0.316 | 4.634 |
| COPSO-BP | 99.594 | 299.469 | 50.082 | 199.092 | 29.937 | 99.501 | 200.481 | 301.800 | 497.670 | -- |
| COPSO-BP relative error(%) | -0.405 | -0.176 | 0.164 | -0.453 | -0.209 | -0.498 | 0.240 | 0.600 | -0.465 | 3.214 |
| NPGA relative error(%) | -6.211 | -0.008 | -0.974 | 3.930 | 3.083 | -0.691 | 0.505 | -2.900 | -3.370 | 19.062 |

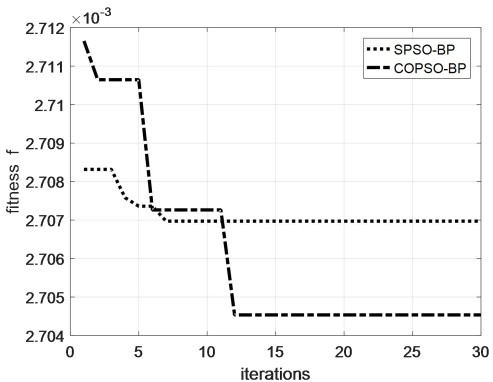 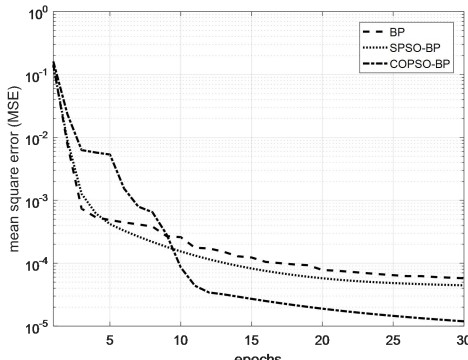

**Fig. 9** Fitness curves of SPSO-BP and COPSO-BP    **Fig. 10** Mean square error curves comparison

**(3) *Inversion comparison***

Three kinds of BP methods as traditional BP, the SPSO-BP and the COPSO-BP algorithms were compared in Table.6. Hence, the training times of COPSO-BP was obviously less than SPSO-BP and was almost equal to BP, it can obtain better precision especially for its global optimization performance.

**Table.6** Simulation comparison for different algorithms

| inversion method | three-layer H type model | | | five-layer KHK type model | | |
|---|---|---|---|---|---|---|
| | training times | minimum training error | test relative error rate(%) | training times | minimum training error | test relative error rate(%) |
| BP | 3 | 0.2882 | 3.284 | 5 | 0.3013 | 5.645 |
| SPSO-BP | 7 | 0.2832 | 2.672 | 15 | 0.2992 | 4.634 |
| COPSO-BP | 5 | 0.2725 | 1.487 | 6 | 0.2900 | 3.214 |

The inversion of COPSO-BP and NGPA were compared in Fig.11. The fitting ability of COPSO-BP was much better than NPGA.

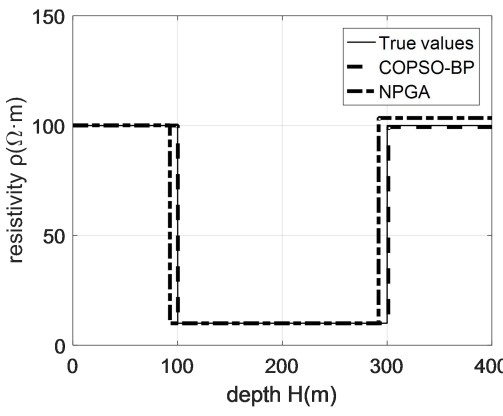 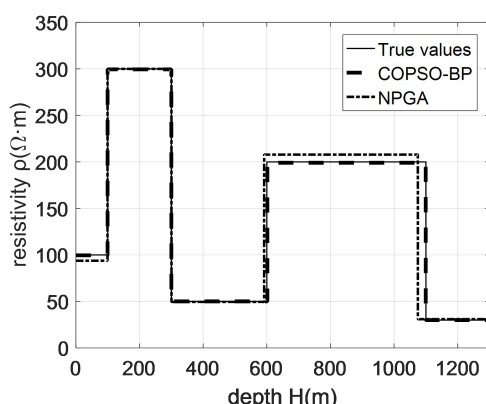

    **(a)** Three-layer H type geoelectric model        **(b)** Five-layer KHK type geoelectric model

**Fig. 11** Inversion comparison for different geoelectric models

**(4) *Robust performance analysis***

In order to verify the algorithm robustness, 5%(26dB) and 10%(20dB) Gaussian random noise
was added in TEM data for three-layer geoelectric model. Three kinds of inversions were
implemented respectively. The results and comparison were shown in Table.7. The $H_z(t)$ and data
with 5% noise were shown in Fig.12.
**Table 7** Comparison of inversion results for three-layer H type (with noise) model

| model parameters | | resistivity $\rho(\Omega \cdot m)$ | | | thickness h(m) | | Total relative error(%) |
|---|---|---|---|---|---|---|---|
| | | $\rho_1$ | $\rho_2$ | $\rho_3$ | $h_1$ | $h_2$ | |
| true value | | 100 | 10 | 100 | 100 | 200 | -- |
| without noise | BP | 99.724 | 9.937 | 100.765 | 99.031 | 198.701 | 3.284 |
| | COPSO-BP | 100.031 | 9.991 | 99.310 | 100.234 | 200.886 | 1.487 |
| 5% noise | BP | 101.374 | 9.966 | 98.283 | 101.255 | 199.282 | 5.039 |
| | COPSO-BP | 100.252 | 9.977 | 98.222 | 101.206 | 199.228 | 3.847 |
| 10% noise | BP | 90.525 | 9.931 | 99.481 | 101.748 | 203.105 | 13.976 |
| | COPSO-BP | 104.472 | 9.96050 | 101.345 | 100.570 | 199.437 | 7.064 |

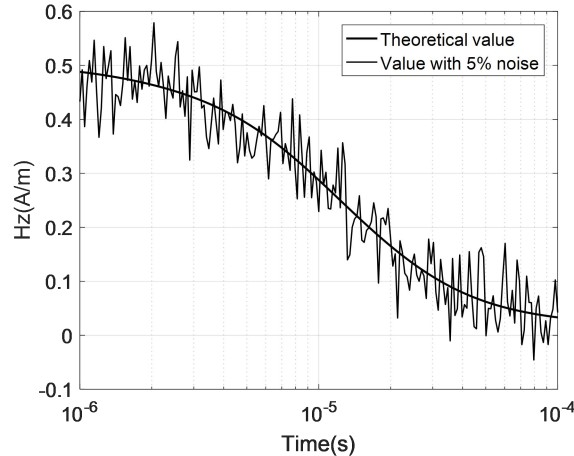

**Fig.12** Forward data of Hz and data with 5% noise
As can be seen from Table 3, after applying 5% and 10% Gaussian noise the COPSO-BP
inversion has higher robust ability. The accuracy was obviously improved based on the total
relative error data.
**4.4 Field example**
In order to test the effectiveness of the method, a transient electromagnetic vertical magnetic field
(Hz) with 10 measuring points at the 380m to 1280m of the No. 1 line from a mining area in
Anhui Province was selected. After the data processing, the inversion was performed using the
3-layer neural network model in the previous section, and the results of BP and COPSOBP
inversion were compared.Figure 13 shows the comparison between the surveyed data and the
inversion data at 380m of the No. 2 line in the mining area.Figure 14 displays the pseudo-sections

of the 10 sets of inversion data combined with the geological data interpolation smoothing.It can be seen from Fig. 14 that the first layer is a low resistivity (100~200 Ω·m), which is inferred to be the second layer (T2g22) gray dolomite of the Middle Triassic old Malague section, with a thickness of about 200 m; the second layer is the second highest resistivity (300~400 Ω·m), which is surmised to be the first layer (T2g21) dolomite of the Middle Triassic old Malaga section, with a thickness of about 400m; the third layer is high resistivity (600~800Ω·m), which is speculated to be the 6th layer (T2g16) limestone dolomite of the Middle Triassic old group. The results are basically consistent with the geological conditions of the mining area, indicating the feasibility and effectiveness of the neural network method.And the results of COPSO-BP inversion are better than those of BP, which the inversion position is more accurate, the shape and spacing are clearer, and the resistivity of each layer is more consistent with the those of the actual geological model.

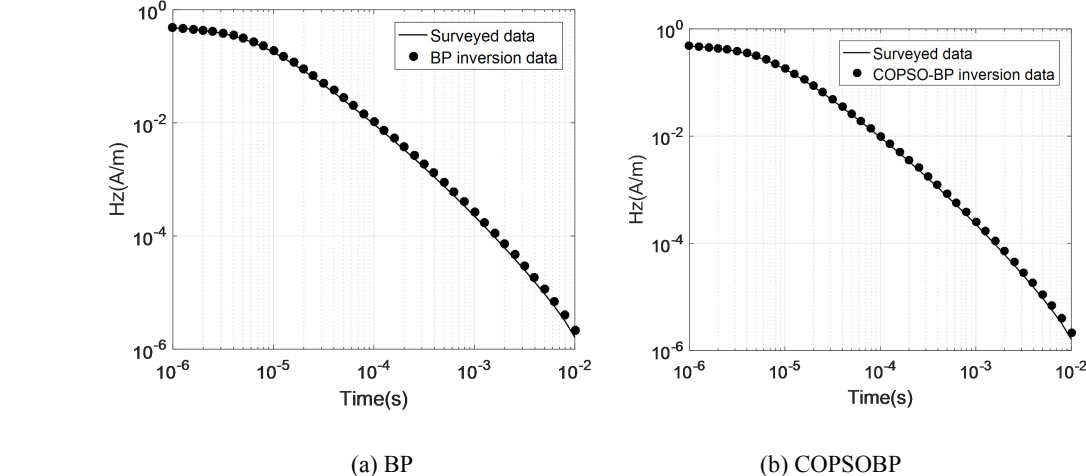

(a) BP                                    (b) COPSOBP

**Figure 13**. 1D inversion forward results. (a) BP; (b) COPSOBP.

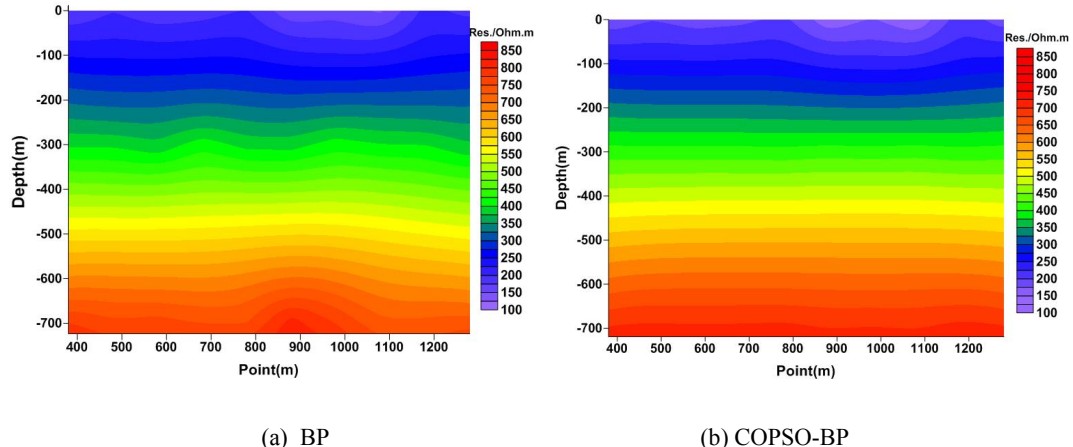

(a) BP                                    (b) COPSO-BP

**Figure 14**. Inversion results of BP (a) and COPSO-BP (b).

## 5 Discussion

The inversion is performed for 3-layered (H-type) and 5-layered (KHK-type) geoelectric models in this paper. The results show that the BP neural network is better than the NPGA algorithm,

because the BP method does not need to use the forward algorithm repeatedly, and its calculation
time is short, which is different from the nonlinear heuristic method based on global space search
solution.
The BP main advantage is that it can interpret the transient electromagnetic sounding results
quickly after training the network. Furthermore, BP algorithm could automatically obtain the
"reasonable rules" between input and output data by learning, and it can adaptively store the
learning content in the network weight, which the BP neural network has the high self-learning
and self-adaptation ability. In addition, the superior simulation results of the test function indicate
that the BP algorithm can approximate any nonlinear continuous function with arbitrary precision,
which means it has strong nonlinear mapping ability; the inversion results of the layered
geoelectric model with uncorrelated noise data prove that the BP algorithm has strong robustness,
which means it has the ability to apply learning results to new knowledge. However, the BP neural
network weight is gradually adjusted by the direction of local improvement, which causes the
algorithm to fall into local extremum, and the weight converges to a local minimum that leads to
the network training failure; Moreover, BP is very sensitive to the initial network weight, and the
initialization network with different weight values tends to converge to different local minimums,
so that obtains different results each time; In addition, the BP algorithm is a gradient descent
method essentially, which leads to a slow convergence rate.
From the results of the layered model and parametric analysis part, it can be seen that single
BP algorithm has higher error value than SPSO-BP, because BP method is sensitive to initial
weight and easy to fall into local minimum values, thus a heuristic global search particle swarm
optimization algorithm with simple structure, rapid convergence and high precision is applied to
optimize the weight and threshold of BP neural network, which improves the global optimization
performance of the algorithm. Furthermore, the PSO algorithm adjusts the inertia weight
adaptively based on the chaotic oscillation curve that is similar to the annealing process in the
simulated annealing algorithm (SA), which jumps out the local extremum faster in the early stage
and accelerates the convergence and reduces the training times. Therefore, compared with
SPSO-BP and BP algorithm, the inversion results of COPSO-BP are closer to the theoretical data
with smaller error fluctuations, stronger anti-noise, better generalization performance and higher
stability, which it is effective in solving geophysical inverse problems.
From the simulation experiment, it is not clear how the weight organization affects the BP
neural network weight learning process. It is necessary to conduct a more systematic study on this
problem to improve our understanding of how BP neural network handles training data.

## 6 Conclusion

The nonlinear COPSO-BP method was proposed for TEM inversion. The BP's initial weight and
threshold parameters were trained by COPSO algorithm which makes it not easy to fall into local
optimum. The chaotic oscillation inertia weight for PSO was proposed so as to improve the PSO's
global optimization ability and fast convergence in early stage. The layered geoelectric model
inversion showed that the COPSO-BP method has better accuracy, stability and relative less
training times.

## 351 Author Contributions

Huaiqing Zhang conceived this manuscript. Ruiyou Li and Huaiqing Zhang developed the main
algorithmic idea and mathematical part. Ruiheng Li and Nian Yu carried out the simulation and
data analysis. Qiong Zhuang completed the writing and interpretation of this manuscript. All
authors contributed to the manuscript writing and approved the final manuscript.

## 357 Competing interests

The authors declare that they have no conflict of interest.

## 360 Acknowledgments

This work was partly supported by the National Natural Science Foundation of China
(No.51377174, No.51577016, No.51877014), the Fundamental Research Funds for the Central
Universities(No.2018CDQYDQ0005).

## 364 Computer Code Availability

Code name is PSOBP, developer is Huaiqing Zhang and Ruiyou Li, contact address is
Chongqing University in China, telephone number is 13752954568 and e-mail is
zhanghuaiqing@cqu.edu.cn, year first available, hardware required is a computer, software
required is MATLAB R2016a, program language is C++, program size is 10KB, and source code
from *https://github.com/liruiyou/PSOBP.*

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
