# Peer review of "A fast approximation for 1D Inversion of Transient"

_Nonlinear Processes in Geophysics, 2019_

## Referee Comment (RC1) · Anonymous Referee #1 · 10 Sep 2019

Ref: npg-2019-36

Title: BP Neural Network and improved Particle Swarm Optimization for Transient Electromagnetic Inversion

Journal: Nonlinear Processes in Geophysics

Dear Editor,

I appreciate the author's effort on the paper. However, the paper requires a revision considering my comments given below.

Comments

1. Page 2, line 53 → Please provide a correct reference name for Fernndez et al. (2010).
2. Page 6, line 136 → There is no $T_k$ and $O_k$ terms in equation 14. In addition, is there no unit for the training error value calculated?
3. Page 7, line 148 → use slightly better instead of litter better.
4. Do you have any experimental studies (i.e., parameter tuning) for the PSO parameters used in the study?
5. The sentence given below requires a reference.
   "*Comparing to the standard PSO (SPSO), a chaotic oscillation inertia weight PSO (COPSO) which can accelerate the convergence rate in the early stage was proposed naturally.*"
   The inertia weight value used in SPSO-BP approach is not clear in the text. Based on my experiments for parameter estimation from geophysical anomalies (e.g., self-potential, gravity, magnetic) using PSO algorithm, the values including 2.041 ($c_1$), 0.948 ($c_2$) and 0.729 ($\omega$) proposed by Carlisle and Dozier (2001) mainly provide quite efficient results. Please provide a comparison.
6. Considering the results presented in Table 2 and Fig.3, is there a possibility to use the same initial population during the evaluation process to provide a good comparison?
7. Please use more proper terms in the text regarding a geophysical optimization study (e.g., predict and desired outputs).
8. Please depict $\omega_s$ and $\omega_e$ inertia weight values in title of Table 3.
9. Use true values instead of reference value and theoretical curve instead of theory curve. In fact, I do not see any curve in Fig. 11. They represent layer parameters.

10. Please define PSO parameter values used in the synthetic case.

11. Please discuss the main advantages and disadvantages of the BP compared to the metaheuristic approaches requiring a parameter space which can be chosen

12. Such a study must include the effect of the noise on the solution in the synthetic case. Besides uncertainty analyses for estimated parameters should be applied for data sets with and without noise. A field example must be also presented.

---

## Referee Comment (RC2) · Anonymous Referee #2 · 20 Sep 2019

Dear Editor,

I reviewed the paper whose title is "BP Neural Network and improved Particle Swarm Optimization for Transient Electromagnetic Inversion" written by Zhang et. al. It is interesting paper. The present form of the paper is not suitable to publish in the journal. However, I have some comments to the authors:

1. The main problem is the TEM forward calculation in this manuscript. It is not clear for me. Is it frequency or time domain? The authors said that this is a transient EM. However, they started derivation with the frequency domain expression using Kaufman's (1983) book, then they obtained Hz(t) response using Gravier –Stehfest method. If you

start a frequency domain, after getting a layered response function you need to get the Fourier transform to get back to in the time domain. Either frequency or time domain we need to use some kind of filter function, since there is no analytic solution for a layered earth. Thus, we use some approximations. In addition, I don't see an apparent resistivity formula in the manuscript. Do they use a late time or early time approximation for the apparent resistivity calculation (or all time approximation)? I would like to see a clear explanation about the apparent resistivity formula and TEM forward response explanation in the manuscript. Please be clear about the TEM forward calculation.

2. There is no field data for the inversion as an example, which is very important. All calculation is synthetic. The manuscript can be published in this journal after my suggestion completed.

Best regards,

(Note: Upon request I can provide a field data set to the Authors. I am running a project; the project includes TEM field measurement. )

---

## Author Comment (AC2) · 28 Sep 2019

The comment was uploaded in the form of a supplement: https://www.nonlin-processes-geophys-discuss.net/npg-2019-36/npg-2019-36-AC2-supplement.zip

---

## Author Comment (AC3) · 28 Sep 2019

The comment was uploaded in the form of a supplement: https://www.nonlin-processes-geophys-discuss.net/npg-2019-36/npg-2019-36-AC3-supplement.zip

---

## Author Comment (AC4) · 28 Sep 2019

The comment was uploaded in the form of a supplement:
https://www.nonlin-processes-geophys-discuss.net/npg-2019-36/npg-2019-36-AC4-supplement.zip

---

## Author Response (AR2)

**Cover letter**

Dear Editor:

On behalf of my co-authors, we thank you very much for giving us an opportunity to revise our manuscript, we appreciate editor and reviewers very much for their positive and constructive comments and suggestions on our manuscript entitled "BP Neural Network and improved Particle Swarm Optimization for Transient Electromagnetic Inversion". (MS No: npg-2019-36).

We have studied reviewer's comments carefully and have made revision which marked in red in the paper. We have tried our best to revise our manuscript according to the comments. Attached please find the revised version, which we would like to submit for your kind consideration.

We would like to express our great appreciation to you and reviewers for comments on our paper. Looking forward to hearing from you.

Thank you and best regards.

Yours sincerely,

Huaiqing Zhang

The State Key Laboratory of Transmission Equipment and System Safety and Electrical New Technology, Chongqing University, Chongqing, China

Tel: + 86 13752954568

E_mail: zhanghuaiqing@cqu.edu.cn

Dear Editors and Reviewers:

  Thank you for your letter and for the reviewers' comments concerning our manuscript entitled "BP Neural Network and improved Particle Swarm Optimization for Transient Electromagnetic Inversion". (MS No.: npg-2019-36). Those comments are all valuable and very helpful for revising and improving our paper, as well as the important guiding significance to our researches. We have studied comments carefully and have made correction which we hope meet with approval. Revised portion are marked in red in the paper. The main corrections in the paper and the responds to the reviewer's comments are as flowing:

For your guidance, itemized response to each review's comments is appended below.

Reviewer #1:

Dear reviewers:

Comments:

1. Page 2, line 53 → Please provide a correct reference name for Fernndez et al. (2010).

2. Page 6, line 136 → There is no T k and O k terms in equation 14. In addition, is there no unit for the training error value calculated?

3. Page 7, line 148 → use slightly better instead of litter better.

4. Do you have any experimental studies (i.e., parameter tuning) for the PSO parameters used in the study?

5. The sentence given below requires a reference.

"Comparing to the standard PSO (SPSO), a chaotic oscillation inertia weight PSO(COPSO) which can accelerate the convergence rate in the early stage was proposed naturally."The inertia weight value used in SPSO-BP approach is not clear in the text. Based on my experiments for parameter estimation from geophysical anomalies (e.g., self-potential, gravity, magnetic) using PSO algorithm, the values including 2.041 ($c_1$ ), 0.948($c_2$ ) and 0.729 ( ) proposed by Carlisle and Dozier (2001) mainly provide quite efficient results. Please provide a comparison.

6. Considering the results presented in Table 2 and Fig.3, is there a possibility to use the same initial population during the evaluation process to provide a good comparison?

7. Please use more proper terms in the text regarding a geophysical optimization study(e.g., predict and desired outputs).

8. Please depicts   and   e inertia weight values in title of Table 3.

9. Use true values instead of reference value and theoretical curve instead of theory curve.
 In fact, I do not see any curve in Fig. 11. They represent layer parameters.

10. Please define PSO parameter values used in the synthetic case.

11. Please discuss the main advantages and disadvantages of the BP compared to the metaheuristic approaches requiring a parameter space which can be chosen

12. Such a study must include the effect of the noise on the solution in the synthetic case.Besides uncertainty analyses for estimated parameters should be applied for data sets with and without noise. A field example must be also presented.

1- Reply:

1-Page 2, line 53 → Please provide a correct reference name for Fernndez et al. (2010).

 (1) We are sincerely sorry for the negligence of the author's name spelling when citing references.

We have made changes and amend them as follows. At the same time, we are also very grateful for your careful review of the manuscript.

Fernández et al. (2010) successfully introduced the PSO in 1-D resistivity inversion.

2-Page 6, line 136 → There is no T k and O k terms in equation 14. In addition, is there no unit for the training error value calculated?

(2) Due to our negligence, '$T_k$' and '$O_k$' terms on line 136 of the page 6 are misspelled and have been modified as follows: $T_s$ , $O_s$ are the expected and predicted output for training sample respectively. Meanwhile, in order to evaluate the effect of model training in this paper, the training error expression is adopted as: $E = \dfrac{1}{S}\sum_{s}^{S}(T_s - O_s)^2$ . In the formula, the training error E only represents the error value, so there is no unit for the training error value calculated.

3-Page 7, line 148 → use slightly better instead of litter better.

(3) According to the meaning in the paper, it is more appropriate that 'slightly better' term than 'litter better', thank you again for your valuable suggestions.

4- Do you have any experimental studies (i.e., parameter tuning) for the PSO parameters used in the study?

(4) In the research of this paper, some experiments are performed on the various different parameter values in the PSO algorithm. The experiment results show that the learning factor (c1 and c2), inertia weight (w), population size (M) and maximum iterations ($T_{max}$) have a little influence on the PSO optimization results.

In PSO, the learning factors $c_1$ and $c_2$ determine the influence of the experience of the particle itself and on the trajectory of the particle group, reflecting the information exchange between the particle swarms. Setting a larger value of $c_1$ will cause the particles to linger too much in the local range, while a larger $c_2$ will cause the particles to converge prematurely to the local minimum, thus setting a larger or smaller $c_1$ and $c_2$ is not conducive to the search of particles. Ideally, the particles should initially search the entire space as much as possible, and at the end of the search, the particles should avoid falling into local extrema. The general setting is $c_1=c_2=2$.

Among the adjustable parameters of the algorithm, the inertia weight is the most significant, which determines the influence of the previous flight speed of the particle on the current flight speed. Therefore, the balance between the global search and the local search can be achieved by adjusting the value of the inertia weight $w$: when $w$ is large, the global search ability is strong, and the local search ability is weak; when $w$ is small, the global search ability is weak, and the local search ability is strong. Thence, the proper inertia weight can improve the optimization ability, and reduce the iterations. However, there is still some difficulty to achieve the optimal performance, because when the inertia weight is large, it is conducive to global search with fast convergence rate, but it is not easy to get an exact solution. When the inertia weight is small, it is beneficial to local search and getting exacter solution, but the convergence is slow and sometimes it falls into local extremum. Therefore, in the PSO optimization algorithm, it is hoped that there will be a higher global search ability in the early stage to find a suitable seed, and a higher development capability in the later stage to accelerate the convergence speed. Thence, the inertial weight is used as a typical linear decreasing strategy. The formula of $w$ is $w(t) = w_{start} - ((w_{start} - w_{end})/T_{max}) \times t$ , in which, according to studies by Y. Shi et al. (1999), the initial value of inertia weight is $w_{start}$=0.9, and the end value of inertia weight is $w_{end}$=0.4,

which can make PSO explore more at the beginning and locate the approximate position of the optimal solution faster. As $w$ gradually decreases, the particle speed slows down and a fine local search begins. This method enables PSO to better control global and local search capability, speed up the convergence and improve the performance of the algorithm. At the same time, in order to further improve the global search ability of the PSO algorithm, the chaotic oscillation inertia weight is adopted as

$$\omega_c(t) = \omega_e + (\omega_s - \omega_e)(0.99^t \cdot x_t), \text{ which } x_{t+1} = \mu x_t(1 - x_t),$$ the experiments results prove

that the best results can be obtained when $x_0 = 0.234$ and $\mu = 4$, which the equation is in a completely chaotic state.

As for the population size (M), the number of individuals contained in the population. In general, there are the higher population diversity and stronger the search ability for algorithm as the population size is larger, therefore, the probability of obtaining an optimal solution is greater, but it also takes more calculation time. Therefore, after a series of numerical experiments, it is reasonable as the population size M=60 and the maximum iteration number Tmax=30.

5-The sentence given below requires a reference.

"Comparing to the standard PSO (SPSO), a chaotic oscillation inertia weight PSO(COPSO) which can accelerate the convergence rate in the early stage was proposed naturally."

The inertia weight value used in SPSO-BP approach is not clear in the text. Based on my experiments for parameter estimation from geophysical anomalies (e.g., self-potential, gravity, magnetic) using PSO algorithm, the values including 2.041 ($c_1$), 0.948($c_2$) and 0.729 (w) proposed by Carlisle and Dozier (2001) mainly provide quite efficient results. Please provide a comparison.

(5) According to your suggestion, the corresponding reference has been added in the corresponding position of the article for the article more rigorous, the literature is as follows:

Shi, X. M., Xiao, M., Fan, J. K., Yang, G. S., and Zhang, X. H.: The damped PSO algorithm and its application for magnetotelluric sounding data inversion, Chinese Journal of Geophysics., 52, 1114−1120, https://doi.org/10.3969/j.issn.0001-5733.2009.04.029, 2009.

We are sorry that the SPSO-BP algorithm inertia weight value is not clearly explained in the manuscript. In this paper, the inertia weight is used as a typical linear decrement strategy, which

the calculation formula (w) is $\omega_1(t) = \omega_s - (\omega_s - \omega_e)t/T_{max}$. At the same time, the definition of

SPSO algorithm inertia weight is supplemented on line 129 of the manuscript. Among them, according to a large number of experimental studies by Y. Shi et al. (1999), the inertia weight initial value (ws) is 0.9, and the inertia weight ending value (we) is 0.4, which can make PSO better control global search ability and local search ability, and speed up the convergence and improve the algorithm performance. The learning factors and inertia weight parameter values in PSO has been elaborated in the previous question (4). In fact, we found that different parameter values have certain influence on the PSO optimization results under different research models by a lot of researches, that is, when the different models achieve the best optimize results using PSO, the various parameter values are different. Therefore, through a series of test functions and geoelectric model inversion experiments, the good optimization results are obtained when the parameter value c1=c2=2, which the effect is better than the results optimized under 2.041 (c1), 0.948 (c2) and 0.729 (w). Among them, the results of SPSO-BP optimization under different parameter values are as follows:

**Table.1** Comparison of the different parameter values in SPSO-BP algorithm for testing functions

| Testing functions | SPSO-BP($c_1=c_2=2$,$w=w_1$) | | SPSO-BP($c_1=2.041$,$c_2=0.948$,$w=0.729$) | |
|---|---|---|---|---|
| | Average value | Optimal value | Average value | Optimal value |
| *Rosenbrock* | 2.375e-3 | 2.300e-5 | 0.3911 | 3.1665e-04 |
| *Bohachevsky* | 0.225 | 1.024e-3 | 0.2832 | 0.0013 |

**Table.2** Inversion comparison of three-layer H type geoelectric model

| H type | resistivity $\rho$ ($\Omega\cdot$m) | | | thickness $h$(m) | | total relative error(%) |
|---|---|---|---|---|---|---|
| | $\rho_1$ | $\rho_2$ | $\rho_3$ | $h_1$ | $h_2$ | |
| reference value | 100 | 10 | 100 | 100 | 200 | -- |
| SPSO-BP relative error(%) ($c_1=c_2=2$,$w=w_1$) | 0.062 | -0.322 | -0.737 | -0.579 | -0.970 | 2.672 |
| SPSO-BP relative error(%) ($c_1=2.041$,$c_2=0.948$,$w=0.729$) | 0.2438 | 2.3154 | -0.566 | 0.9707 | -0.3327 | 4.4290 |

6- Considering the results presented in Table 2 and Fig.3, is there a possibility to use the same initial population during the evaluation process to provide a good comparison?

(6) According to your suggestion, the same initialized population is used for the test function optimization. The search curves for the Rosenbrock and Bohachevsky test functions are shown below fig.1, and made changes in the manuscript. However, the comparative study results show that the same initial population has no significant effect on the final optimization results, so that it is almost negligible.

[Figure]

**Fig. 1** Training error curves of SPSO-BP and COPSO-BP algorithms

7-Please use more proper terms in the text regarding a geophysical optimization study(e.g., predict and desired outputs).

(7) Thank you for your suggestion. For the rigor of the article, we have modified the inappropriate terminology, such as 'predict output' modified to 'predict value', 'desired output' modified to 'actual value', 'reference value' modified to 'true values'.

8-Please depict $w_s$ and $w_e$ inertia weight values in title of Table 3.

(8) In order to reasonably compare the effects of four different inertia weights in the PSO algorithm, the same initial inertia weight value and end inertia weight value are used in this paper, such as inertia weight $\omega s = 0.9$, $\omega e = 0.4$. At the same time, based on the valuable comments made by the reviewers, in order to facilitate the reader to more clearly understand the comparison effect for

different inertia weights, we have refined the title of Table 3 as follows:

**Table.3** Comparison of different inertia weights in PSO algorithms ( $\omega_s$ = 0.9, $\omega_e$ = 0.4).

9-Use true values instead of reference value and theoretical curve instead of theory curve.

In fact, I do not see any curve in Fig. 11. They represent layer parameters.

(9) Thank you very much for pointing out the incorrect terminology in the paper and giving appropriate revisions. We have made corresponding revisions, such as the 'reference value' in Table 4 and Table 5 is modified to 'true values', the 'theory curve' is modified to 'theoretical curve'.

As you can see, the theoretical curve in Figure 2 of the first draft does represent the layered parameter value. However, due to the inappropriate naming of the line segments in the figure, it is inconvenient for the reader to understand, so after careful consideration, the 'theory curve' is modified to 'True values' as shown below.

[Figure]

(a) Three-layer H type geoelectric model  (b) Five-layer KHK type geoelectric model

**Fig. 2** Inversion comparison for different geoelectric models

10- Please define PSO parameter values used in the synthetic case.

(10)Due to our negligence, it not clearly account for the various parameters values of PSO algorithm in the Layered model and parameter analysis part. However, in fact, the PSO parameter value in this synthetic case is the same as the PSO parameter value of Algorithm Testing part. Therefore, the following is added to Section 4.3 of this paper.

A 3-layered and 5-layered geoelectric models were investigated, which the PSO parameter values are the same as those of the Algorithm Testing parts in the paper.

11-Please discuss the main advantages and disadvantages of the BP compared to the metaheuristic approaches requiring a parameter space which can be chosen.

(11)Regarding the advantages and disadvantages of BP compared to the metaheuristic approaches requiring a parameter space which can be chosen, which has been elaborated in the paper discussion part, and its detailed description is as follows:

At present, heuristic algorithms are mainly based on natural body algorithms, which mainly includes ant colony algorithm, simulated annealing method, particle swarm optimization, ant optimization, fish swarm algorithm, bee colony algorithm and so on. And heuristic algorithms have a common feature: starting from a random feasible initial solution, an iterative improvement strategy is adopted to approximate the optimal solution of the problem. The advantage of the heuristic algorithm is that it is more efficient than the blind search method. In addition, a carefully designed heuristic function often gets the optimal solution in a very short time. However, the heuristic algorithm needs to

repeatedly call the forward algorithm for each iteration in nonlinear resistivity inversion, resulting in a long calculation time.

However, BP neural network is the most active branch for nonlinear resistivity inversion. The inversion algorithm is different from the nonlinear heuristic method based on global solution space search, which it does not need to call the forward algorithm repeatedly, so its calculation time is short. At the same time, BP can approximate any nonlinear continuous function with arbitrary precision, which makes BP have strong nonlinear mapping ability. In addition, BP neural network can automatically extract "reasonable rules" between output and output data through learning, and adaptively memorize the learning content in the weight of the network, that is, BP neural network has the ability of high self-learning and self-adaptation; and BP neural network can be trained after training mode or noise-contaminated mode for correct prediction, that is, BP neural network has the generalization ability to apply learning results to new knowledge; in addition, the BP neural network does not have a great impact on the global training results after the local or part of the neurons are destroyed, that is, the BP neural network has certain fault tolerance.

However, the weight of the BP neural network is gradually adjusted by the direction of local improvement, which causes the algorithm to fall into local extremum, and the weight converges to the local minimum point, which leads to network training failure; in addition, BP is very sensitive to the initial network weights, initializing the network with different weights, which tends to converge to different local minima, resulting in different results for each training; and BP algorithm is essentially a gradient descent method, resulting in BP has a slow convergence rate; and the approximation and generalization ability of the BP neural network model is closely related to the learning samples, that is, the BP neural network depends on the sample selection. In view of the fact that the neural network is sensitive to the initial weight and easy to fall into the local minimum, the heuristic global search particle swarm optimization algorithm with simple structure, fast convergence and high precision is used to optimize the initial weight and threshold of the neural network. The method is stable and effective, and is not easy to fall into local optimum, and has better performance.

12-Such a study must include the effect of the noise on the solution in the synthetic case.Besides uncertainty analyses for estimated parameters should be applied for data sets with and without noise. A field example must be also presented.

(12) Anti-noise tests and a field example are added as follows. Since the result of the inversion is affected by various control parameters, computer system performance and programming planning in the algorithm, parameter uncertainty analysis should be performed in the paper, which it is the insufficiency of the research work, and it is also the direction we need further research. We sincerely hope to get your understanding.

(4) *Robust performance analysis*

In order to verify the algorithm robustness, 5%(26dB) and 10%(20dB) Gaussian random noise was added in TEM data for three-layer geoelectric model. Three kinds of inversions were implemented respectively. The results and comparison were shown in Table.3. The $H_z(t)$ and data with 5% noise were shown in Fig.3.

**Table 3** Comparison of inversion results for three-layer H type (with noise) model

| model | resistivity $\rho$($\Omega$•m) | thickness h(m) | Total relative |
| --- | --- | --- | --- |

| parameters | | $\rho_1$ | $\rho_2$ | $\rho_3$ | $h_1$ | $h_2$ | error(%) |
|---|---|---|---|---|---|---|---|
| true value | | 100 | 10 | 100 | 100 | 200 | -- |
| without noise | BP | 99.724 | 9.937 | 100.765 | 99.031 | 198.701 | 3.284 |
| | COPSO-BP | 100.031 | 9.991 | 99.310 | 100.234 | 200.886 | 1.487 |
| 5% noise | BP | 101.374 | 9.966 | 98.283 | 101.255 | 199.282 | 5.039 |
| | COPSO-BP | 100.252 | 9.977 | 98.222 | 101.206 | 199.228 | 3.847 |
| 10% noise | BP | 90.525 | 9.931 | 99.481 | 101.748 | 203.105 | 13.976 |
| | COPSO-BP | 104.472 | 9.96050 | 101.345 | 100.570 | 199.437 | 7.064 |

[Figure]

**Fig.3** Forward data of Hz and data with 5% noise

As can be seen from Table 3, after applying 5% and 10% Gaussian noise the COPSO-BP inversion has higher robust ability. The accuracy was obviously improved based on the total relative error data.

**4.4 Field example analysis**

In order to test the effectiveness of the method, a transient electromagnetic vertical magnetic field (Hz) with 10 measuring points at the 380m to 1280m of the No. 1 line from a mining area in Anhui Province was selected. After the data processing, the inversion was performed using the 3-layer neural network model in the previous section, and the results of BP and COPSOBP inversion were compared.Figure 4 shows the comparison between the surveyed data and the inversion data at 380m of the No. 2 line in the mining area.Figure 6 displays the pseudo-sections of the 10 sets of inversion data combined with the geological data interpolation smoothing.It can be seen from Fig. 5 that the first layer is a low resistivity (100~200 Ω·m), which is inferred to be the second layer (T2g22) gray dolomite of the Middle Triassic old Malague section, with a thickness of about 200 m; the second layer is the second highest resistivity (300~400 Ω·m), which is surmised to be the first layer (T2g21) dolomite of the Middle Triassic old Malaga section, with a thickness of about 400m;the third layer is high resistivity (600~800Ω·m), which is speculated to be the 6th layer (T2g16) limestone dolomite of the Middle Triassic old group.The results are basically consistent with the geological conditions of the mining area, indicating the feasibility and effectiveness of the neural network method.And the results of COPSO-BP inversion are better than those of BP, which the inversion position is more accurate, the shape and spacing are clearer, and the resistivity of each layer is more consistent with the those of the actual geological model.

[Figure]

(a) BP          (b) COPSOBP

Figure 4. 1D inversion forward results. (a) BP; (b) COPSOBP.

[Figure]

(a) BP          (b) COPSO-BP

Figure 5. Inversion results of BP (a) and COPSO-BP (b).

Special thanks to you for your good comments.

Reviewer #2:

Dear reviewers:

Comments:

1. The main problem is the TEM forward calculation in this manuscript. It is not clear for me. Is it frequency or time domain? The authors said that this is a transient EM. However, they started derivation with the frequency domain expression using Kaufman's (1983) book, then they obtained Hz(t) response using Gravier –Stehfest method. If you start a frequency domain, after getting a layered response function you need to get the Fourier transform to get back to in the time domain. Either frequency or time domain we need to use some kind of filter function, since there is no analytic solution for a layered earth. Thus, we use some approximations. In addition, I don't see an apparent resistivity formula in the manuscript. Do they use a late time or early time approximation for the apparent resistivity calculation (or all time approximation)? I would like to see a clear explanation about the apparent resistivity formula and TEM forward response explanation in the manuscript. Please be clear about the TEM forward calculation.

2. There is no field data for the inversion as an example, which is very important. All calculation is

synthetic. The manuscript can be published in this journal after my suggestion completed.

2- Reply:

1- The main problem is the TEM forward calculation in this manuscript. It is not clear for me. Is it frequency or time domain? The authors said that this is a transient EM. However, they started derivation with the frequency domain expression using Kaufman's (1983) book, then they obtained Hz(t) response using Gravier –Stehfest method. If you start a frequency domain, after getting a layered response function you need to get the Fourier transform to get back to in the time domain. Either frequency or time domain we need to use some kind of filter function, since there is no analytic solution for a layered earth. Thus, we use some approximations. In addition, I don't see an apparent resistivity formula in the manuscript. Do they use a late time or early time approximation for the apparent resistivity calculation (or all time approximation)? I would like to see a clear explanation about the apparent resistivity formula and TEM forward response explanation in the manuscript. Please be clear about the TEM forward calculation.

(1)We are sincerely sorry that the TEM forward calculation has not been explained clearly in the manuscript. After careful consideration and modification, the relevant content has been added in the Forward Model part of article and was elaborated as follows:

**一. TEM forward response explanation**

The forward model of this paper belongs to the time domain. Due to the high-frequency oscillation and slow decay characteristics of the Bessel function in the formula (15) of this manuscript, the analytical solution can only be obtained in uniform half-space. However, the layered geoelectric model can only be solved by numerical calculation method.

Considering the Hankel transformation of the first-order Bessel function, Anderson used a linear numerical filtering method with a filter coefficient of 283, which achieved better results.Then based on Anderson's research, D.Guptasarma and B.Singh (1997) improved the digital filtering algorithm and gave filter coefficients which of 61 and 120 points J0, 47 and 140 points J1, and getting higher calculation accuracy. This paper adopts the improved digital filtering method, the filter coefficient of 47 points J1 is selected by a large number of experiments, so that the frequency domain response of layered earth with the center loop source can be obtained. Then, according to the Laplace transform and the related properties, the frequency domain response is converted into a complex frequency domain. Next, due to the accuracy of the Gaver-Stehfest algorithm is higher than that of the Guptasarma algorithm in the late stage, and the method has the characteristics of slow filtering speed and pure real number operation, which makes it faster than cosine transform, so that the algorithm can be used to solve complex terrains. Finally, the Gaver-Stehfest algorithm is selected to transform the complex frequency domain response into a time domain response. Among them,the 12-point filter coefficient is used in the G-S transform algorithm, and the detailed derivation process and explanation are as follows:

**1)Frequency domain response of the center loop source**

For circular emission loops, the analytical solution is typically derived from the area of the perpendicular magnetic dipole source or the line integral of the horizontal magnetic dipole. Consider the vertical dipole $dm$( $dm = \mathrm{I}\rho d\varphi d\rho$ ), and the emission current I. Calculate the area of $dm$ along the entire loop through a surface integral:

$$A_z^* = \frac{I}{2\pi} \int_0^\infty \frac{m}{m + m_1/R_1^*} \int_0^a \int_0^{2\pi} J_0\left(mR\right)\rho' d\varphi d\rho' \mathrm{d}m \tag{1}$$

Using the following formula:

$$J_0(mR) = \sum_{n=-\infty}^{\infty} J_n(m\rho)J_n(m\rho')\cos n\varphi \tag{2}$$

Bring into equation (1) and interchange the integral with the summation order:

$$A_z^* = \frac{I}{2\pi}\int_0^\infty \frac{m}{m+m_1/R_1^*}\sum_{n=-\infty}^{\infty} J_n(m\rho)\int_0^a\int_0^{2\pi} J_n(m\rho')\rho'\cos n\varphi d\varphi d\rho' dm \tag{3}$$

In the above formula, the inner layer integral is not zero only when $n=0$, so there are:

$$A_z^* = I\int_0^\infty \frac{m}{m+m_1/R_1^*}J_0(m\rho)\int_0^a J_0(m\rho')\rho' d\rho' dm \tag{4}$$

Using the below relation:

$$\int x^n J_{n-1}(x)dx = x^n J_n(x) \tag{5}$$

We can obtain:

$$A_z^* = Ia\int_0^\infty \frac{1}{m+m_1/R_1^*}J_0(m\rho)J_1(ma)dm \tag{6}$$

Similarly, we can get:

$$H_z = Ia\int_0^\infty \frac{m^2}{m+m_1/R_1^*}J_0(m\rho)J_1(ma)dm \tag{7}$$

where $a$ is the radius of the circular transmitting coil.

**Therefore, the frequency domain response of the center loop ( $\rho=0$) can be acquired as:**

$$H_z = Ia\int_0^\infty \frac{m^2}{m+m_1/R_1^*}J_1(ma)dm \tag{8}$$

Among them, the recursive relationship can be gained after a series of derivations:

$$\begin{cases} R_n^* = 1 \\ R_j^* = \dfrac{m_j R_{j+1}^* + m_{j+1}\text{th}\left(m_j h_j\right)}{m_{j+1} + m_j R_{j+1}^*\text{th}\left(m_j h_j\right)} \quad j = n-1, n-1, \cdots, 1 \end{cases} \tag{9}$$

① **Analytic calculation of frequency domain response in uniform half space**

Through the above formula, the frequency domain response of the central return line ($n=1$,

$R_1^* = 1$) in a uniform half space can be obtained as follows:

$$H_z = Ia\int_0^\infty \frac{m^2}{m+m_1}J_1(ma)dm \tag{10}$$

Also, we can gain the analytical formula of the vertical magnetic field :

$$H_z = \frac{Ia}{k^2} \int_0^\infty m^2(m-m_1)J_1(ma)dm$$

$$= -\frac{Ia}{k^2}\frac{\partial}{\partial a}\left[\frac{\partial^2}{\partial z^2}\left(\frac{e^{-ik_1}R}{R} - \frac{1}{R}\right)\right] \quad (11)$$

$$= \frac{I}{-k_1^2 a^3}\left[3-(3+3ik_1a - k_1^2 a^2)e^{-ik_1 a}\right]$$

② **Numerical calculation of frequency domain response for layered geoelectric model (Hankel transform)**

In the integral formula (7), due to the high-frequency oscillation and slow decay of the Bessel function, the analytical solution can be obtained only in the uniform half-space. But for the solution of the layered earth, only the numerical calculation method can be used. Anderson used linear numerical filtering, using 283 as the filter coefficient, and achieved ideal results.

Based on Anderson's research, D.Guptasarma and B.Singh (1997) improved the digital filtering algorithm and gave the filter coefficients of 61 and 120 points J0 and the filter coefficients of 47 and 140 points J1. Its calculation accuracy is higher. The calculation formula for the improved digital filtering method of Guptasarma and Singh is:

$$f(r) = \frac{1}{r}\sum_{i=1}^{n} K(\lambda_i)W_i \quad (12)$$

In the formula $\lambda_i = 1/r \times 10[a+(i\text{-}1)s]$, $i = 1, 2, \cdots, n$, $W_i$ is the filter coefficient. After the numerical simulation experiment, the 47-point J1 filter coefficient is selected, so that the frequency domain response of the layered earth transient electromagnetic can be obtained, which lays a foundation for the solution of the next time domain response.

2) **Time domain response of center loop source**

According to the Laplace transform and related properties, the correlation complex frequency domain response function can be obtained, and then The supply current I(s) is multiplied by the vertical magnetic field frequency domain response function $H_z(s)$ to obtain the time domain response $h_z(t)$ and time partial derivative $\frac{\partial h_z(t)}{\partial t}$:

$$\begin{cases} h_z(t) = L^{-1}[I(s)H_z(s)] \\ \dfrac{\partial h_z(t)}{\partial t} = L^{-1}[sI(s)H_z(s)] \end{cases} \quad (13)$$

If the supply current is a unit positive step response, its corresponding Lagrangian transformation is $I(s) = \dfrac{1}{s}$; if the supply current is a unit step response:

$$I(t) = \begin{cases} 0, & t < 0 \\ \dfrac{1}{T_1}t, & 0 \le t < T_1 \\ 1, & T_1 < t \end{cases} \quad (14)$$

Where $T1$ is the off time and its corresponding Lagrangian transformation is

$$I(s) = \frac{1}{T_1}\frac{1}{s^2} - \frac{1}{T_1}\frac{1}{s^2}e^{-T_1 s} = \frac{1}{T_1}\frac{1}{s^2}(1-e^{-T_1 s}) \quad (15)$$

In actual exploration work, to avoid the electromagnetic noise generated by the sky power, the detecting device usually observes the signal induced voltage:

$$V_z(t) = -Sn\mu_0 \frac{\partial h_z(t)}{\partial t} \tag{16}$$

where, $S$ is the area of the receiving coil, and $n$ is the number of turns of the coil. The induced voltage is proportional to the rate of change of the vertical component of the magnetic induction.

① **Analytic calculation of time domain response in uniform half space**

According to the formula (11) and (13), the positive step response of the central loop can be obtained:

$$h_z = \frac{I}{2a}[\frac{3}{2\theta^2 a^2} erf(\theta a) + erfc(\theta a) - \frac{3}{\theta a\sqrt{\pi}} e^{-\theta^2 a^2}] \tag{17}$$

According to $f\text{-}(t)=f(\infty)\text{-}f(t)$, $t>0$, the negative step response of the center loop and the time derivative of the vertical magnetic field can be obtained:

$$h_z = \frac{I}{2a}\left[ \frac{3}{\sqrt{\pi}\theta a} e^{-\theta^2 a^2} + \left(1 - \frac{3}{2\theta^2 a^2}\right) erf(\theta a) \right] \tag{18}$$

$$\frac{\partial h_z}{\partial t} = \frac{I}{\mu_0 \sigma a^3}\left[ 3erf(\theta a) - \frac{2\theta a}{\sqrt{\pi}}\left(3 + 2\theta^2 a^2\right) e^{-\theta^2 a^2} \right] \tag{19}$$

② **Numerical calculation of time domain response of geoelectric model (G-S transformation)**

Through the above, the numerical solution of the frequency domain response of the layered geoelectric model is gained, and then the time domain response is got by frequency-time domain conversion. At present, the main frequency-time domain conversion methods include Fourier transform method, delay spectrum method, Guptasarma filtering algorithm and Gaver-Stehfest inverse Lagrangian transform method. The four methods have their own advantages, disadvantages and applicable scope. Although the Fourier transform method can be used for a variety of geoelectric models and launchers, multiple frequency and kernel function samplings make the calculations large and computationally slow. The delay spectrum method is also called the attenuation spectrum method. The required calculation frequency is small and the calculation amount is small, but the generalized solution of the ill-conditioned matrix needs to be solved, and the late transient response is not stable enough.The late response of the Guptasarma filtering algorithm is more stable than the delay spectrum method, but the number of frequency samples required is increased and is only suitable for calculating a simple geoelectric model.The Guptasarma filtering algorithm is large, but Gaver-Stehfest algorithm has the advantages of pure real number operation, high calculation precision, less required frequency points and can be used to calculate complex geoelectric models. It has wider applicability in electromagnetic detection.

**The complexity of the frequency domain electromagnetic field makes it difficult to solve the time domain theoretical response by Fourier transform method, but the time domain response can be numerically calculated by Gaver-Stehfest inverse Lagrangian transform (G-S transform). Therefore, this paper selects the Gaver-Stehfest algorithm to realize the time-frequency conversion of the central loop.**

Through a series of derivation and conversion simplification, we can get the Gaver algorithm formula:

$$f(t) = \frac{\ln 2}{t} \sum_{i=0}^{n} \frac{(2n)!(-1)^i}{(n-1)!i!(n-i)!} F\left[(n+i)\frac{\ln 2}{t}\right] \tag{20}$$

Stehfest improved the above equation using an interpolation formula whose calculation expression is:

$$f(t) = \frac{\ln 2}{t} \sum_{m=1}^{N} K_m F(s_m) \tag{21}$$

where $s_m=(ln2/t)\times m$, $Km$ is the filter coefficient of the G-S transform algorithm:

$$K_m = (-1)^{m+N/2} \sum_{k=[(m+1)/2]}^{\min(m,N/2)} \frac{k^{N/2}(2k)!}{(N/2-k)!k!(k-1)!(m-k)!(2k-m)!} \tag{22}$$

where $k$ is the integer part of $[(m+1)/2]$ and $N$ is digital length of the computer. At a certain time, the time domain transient response $hz$(t) is the sum of the product of the selected discrete $F(s_m)$ and the coefficient $Km$, thus realizing the frequency domain to time domain conversion. Therefore, the Gaver-Stehfest algorithm is chosen to transform the central loop frequency domain response into a time domain response, in which 12-point filter coefficients are used in the G-S transform algorithm.

**二. The apparent resistivity problem**

Since the vertical magnetic field response Hz is used as input value to the neural network in the manuscript, the apparent resistivity formula is not listed in detail. Regarding the magnetic field strength and apparent resistivity, the method of defining all time apparent resistivity using the magnetic field strength (or magnetic induction) can better reflect the geoelectric model. Among them, when the magnetic resistivity time partial derivative or induced electromotive force is used to define the all time apparent resistivity, multiple solutions or no solutions will occur, and there are obvious false extremums. This method blurs the correct reflection of the formation parameters.The all time apparent resistivity defined by the magnetic field strength or the magnetic induction intensity is a single-valued function, and there is no false extremum. Therefore, the method of defining the apparent resistivity by using the magnetic field strength (or magnetic induction) can better reflect the geoelectric model. It is also meaningful to convert the induced voltage measured in the survey into a vertical magnetic field.Its detailed description of the all time apparent resistivity is as follows:

**1) Definition of apparent resistivity of the central loop**

At present, there are many methods for defining the all time apparent resistivity of the transient electromagnetic method. According to the data source, it can be roughly divided into two types: based on the magnetic field strength (magnetic induction intensity) and based on the induced electromotive force. The transient response($h_z$、$\partial hz/\partial t$) at the center of the center loop in a uniform half-space is as shown in equations (2.51) and (2.52):

$$u = \theta a = \frac{a}{2}\sqrt{\frac{\mu_0}{\rho t}} \tag{23}$$

The expression of the magnetic induction and its time partial derivative is:

$$B_z = \frac{I\mu_0}{2a}\left[\frac{3}{\sqrt{\pi}u}e^{-u^2} + (1-\frac{3}{2u^2})erf(u)\right] \tag{24}$$

$$\frac{\partial B_z}{\partial t} = \frac{I\rho}{a^3}[3erf(u) - \frac{2u}{\sqrt{\pi}}(3+2u^2)e^{-u^2}]\tag{25}$$

The all time apparent resistivity can be expressed as:

$$\rho = \frac{a^2\mu_0}{4tu^2}\tag{26}$$

When using the magnetic induction time partial derivative or the induced electromotive force to define the all time apparent resistivity, multiple solutions or no solutions will occur, and there are obvious false extremums. This method blurs the correct reflection of the formation parameters, and the magnetic field strength or The full-area apparent resistivity defined by magnetic induction is a single-valued function, and there is no false extremum. Therefore, it is better to use the magnetic field strength (or magnetic induction) to define the all time apparent resistivity.

**2) Method of converting induced voltage into vertical magnetic field**

From the above analysis, it can be seen that the all time apparent resistivity defined by the magnetic field strength can better reflect the geoelectric model, therefore, it is very meaningful to convert the induced voltage measured into a vertical magnetic field in the exploration. The relationship between the center loop induced voltage Vz (t) measured by the transient instrument and the vertical

magnetic field Hz(t) is:
$$\frac{\partial H_z(t)}{\partial t} = -\frac{1}{Sn\mu_0}V_z(t)\tag{27}$$

Integrate on both sides of equation (27):

$$\int_a^b \frac{\partial H_z(t)}{\partial t}dt = -\frac{1}{Sn\mu_0}\int_a^b V_z(t)dt\tag{28}$$

When the upper limit of the integral is the time variable t or the lower limit of the integral takes the time variable t, the above formula can be changed to:

$$H_z(t) = \int_a^t \frac{\partial H_z(t)}{\partial t}dt + H_z(a) = -\frac{1}{Sn\mu_0}\int_a^t V_z(t)dt + H_z(a)\tag{29}$$

$$H_z(t) = -\int_t^b \frac{\partial H_z(t)}{\partial t}dt + H_z(b) = \frac{1}{Sn\mu_0}\int_t^b V_z(t)dt + H_z(b)\tag{30}$$

If *Hz* is calculated by equation (29), $H_z(a)$ at time $a \to 0$ is required, but due to the existence of turn-off time, an error will inevitably occur in the calculation, and as time increases, the relative error will continue. Increases will cause late response distortion. It can be seen from the response curve that when the time t is large, the response values of the vertical magnetic field and its time partial derivative tend to be zero, so the response value of the last sampling point can be replaced by 0. When b is large in the formula (30), $H_z(b)=0$, when t decreases continuously, the value of *Hz(t)* increases continuously, and the relative error also decreases. The error for early calculation can be neglected. The calculation formula is:

$$H_z(t) = \frac{1}{Sn\mu_0}\int_t^b V_z(t)dt \qquad (31)$$

2-There is no field data for the inversion as an example, which is very important. All calculation is synthetic. The manuscript can be published in this journal after my suggestion completed.
(Note: Upon request I can provide a field data set to the Authors. I am running a project; the project includes TEM field measurement. )

(2) Thank you very much for your comments, a field example are added as follows. At the same time, we sincerely hope to get your TEM measured data set, we will be particularly grateful.

**4.4 Field example analysis**

In order to test the effectiveness of the method, a transient electromagnetic vertical magnetic field (Hz) with 10 measuring points at the 380m to 1280m of the No. 1 line from a mining area in Anhui Province was selected. After the data processing, the inversion was performed using the 3-layer neural network model in the previous section, and the results of BP and COPSOBP inversion were compared.Figure 4 shows the comparison between the surveyed data and the inversion data at 380m of the No. 2 line in the mining area.Figure 6 displays the pseudo-sections of the 10 sets of inversion data combined with the geological data interpolation smoothing.It can be seen from Fig. 5 that the first layer is a low resistivity (100~200 Ω·m), which is inferred to be the second layer (T2g22) gray dolomite of the Middle Triassic old Malague section, with a thickness of about 200 m; the second layer is the second highest resistivity (300~400 Ω·m), which is surmised to be the first layer (T2g21) dolomite of the Middle Triassic old Malaga section, with a thickness of about 400m;the third layer is high resistivity (600~800Ω·m), which is speculated to be the 6th layer (T2g16) limestone dolomite of the Middle Triassic old group.The results are basically consistent with the geological conditions of the mining area, indicating the feasibility and effectiveness of the neural network method.And the results of COPSO-BP inversion are better than those of BP, which the inversion position is more accurate, the shape and spacing are clearer, and the resistivity of each layer is more consistent with the those of the actual geological model.

[Figure]

(a) BP                    (b) COPSOBP

Figure 4. 1D inversion forward results. (a) BP; (b) COPSOBP.

[Figure]

(b)  BP                         (b) COPSO-BP

Figure 5. Inversion results of BP (a) and COPSO-BP (b).

Special thanks to you for your good comments.

We tried our best to improve the manuscript and made some changes in the manuscript. These changes will not influence the content and framework of the paper. And here we did not list the changes but marked in red in revised paper.

We appreciate for Editors/Reviewers' warm work earnestly, and hope that the correction will meet with approval.

Once again, thank you very much for your comments and suggestions.

---

## Author Response (AR3)

**Response letter**

Dear Editor:

Firstly, we are very sincerely sorry to have uploaded the wrong version in the previous submission. We have now uploaded the newly manuscript and response letter. On behalf of my co-authors, we thank you very much for giving us an opportunity to revise our manuscript, we appreciate editor and reviewers very much for their positive and constructive comments and suggestions on our manuscript entitled "A fast approximation for 1D Inversion of Transient Electromagnetic Data by BP Neural Network and improved Particle Swarm Optimization". (MS No: npg-2019-36).

We have studied reviewer's comments carefully and have made revision which marked in red in the paper. We have tried our best to revise our manuscript according to the comments. Attached please find the revised version, which we would like to submit for your kind consideration.

We would like to express our great appreciation to you and reviewers for comments on our paper. Looking forward to hearing from you.

Thank you and best regards.

Yours sincerely,

Huaiqing Zhang

The State Key Laboratory of Transmission Equipment and System Safety and Electrical New Technology, Chongqing University, Chongqing, China

Tel: + 86 13752954568

E_mail: zhanghuaiqing@cqu.edu.cn

Dear Editors and Reviewers:

Thank you for your letter and for the reviewers' comments concerning our manuscript entitled "A fast approximation for 1D Inversion of Transient Electromagnetic Data by BP Neural Network and improved Particle Swarm Optimization". (MS No: npg-2019-36). Those comments are all valuable and very helpful for revising and improving our paper, as well as the important guiding significance to our researches. We have studied comments carefully and have made correction which we hope meet with approval. Revised portion are marked in red in the paper. The main corrections in the paper and the responds to the reviewer's comments are as flowing:

For your guidance, itemized response to each review's comments is appended below.

Dear Norbert Marwan:

Comments:

1)The newly uploaded manuscript and response letter is the same as for the previous version. Perhaps the authors have uploaded the wrong version?

(1)We are very sincerely sorry to have uploaded the wrong version in the previous submission. We have now uploaded the newly manuscript and response letter, and apologize for any inconvenience again.

Reviewer #1:

Dear reviewers:

Comments:

1) In some sentences, the use of geoelectrical parameters instead of geoelectric parameters can be more suitable. Please also depict in the introduction of the paper what are these parameters. For instance, …geoelectrical parameters consisting of the resistivities and thicknesses of the layers…

2) Based on equation 14 in the text, Ts and Os are unitless.

3) Considering the y-label of Fig. 4, I can not see any explanations related to the Rastrigin benchmark function used in your theoretical case. It would be nice if you also provided a clear version of the manuscript.

4) You declare that Fig. 3 shows one of the evolutionary training error curves. Considering the sentence given below, you should clearly verify whether this finding is also valid for each independent run of the algorithms.

"However, the COPSO-BP has better convergence rate and optimization efficiency in the early stage in Fig.3."

5) Please update Fig. 12 by adding the noise content to the theoretical curve.

6) How did you obtain error values less than the noise content added to the theoretical data?

7) Considering Fig. 14 that present the results of two approaches, the same resistivity range and contour interval must be used.

8) You should also mention in the paper which model parameters did you obtain through the PSO part of the algorithm at the end of 30 iterations. What are their error values as initial estimates requiring by the BP approach?

9) The title of the paper should be reconsidered by the authors. What does it mean "Transient Electromagnetic Inversion". Think about "A fast approximation for 1D Inversion of Transient

Electromagnetic Data by ..” or etc.

1- Reply:
1-In some sentences, the use of geoelectrical parameters instead of geoelectric parameters can be more suitable. Please also depict in the introduction of the paper what are these parameters. For instance, …geoelectrical parameters consisting of the resistivities and thicknesses of the layers…

(1) Special thanks to your great comments on this manuscript. As you can see, some of the terminology as 'geoelectric parameters' that used in the paper is not very suitable. We have already made changes in the manuscript, some of which are modified as follows:

All the geoelectrical parameters and the forward model relations are implied in the weight and threshold parameters of ANN.

For BP structure, the output nodes are determined by the number of inversion geoelectrical parameters,......

The BP training samples which is a series of $H_z(t)$ for different geoelectrical parameters were generated by TEM forward model.

In addition, in order to facilitate the reader's understanding, the geoelectric parameters involved in the introduction of the paper have been elaborated accordingly, as follows:

Transient electromagnetic (TEM) method applies the secondary receiving voltage induced by the rapid switching off pulse current, and then deduces the geoelectrical parameters consisting of the resistivities and thicknesses of the layers.

2-Based on equation 14 in the text, Ts and Os are unitless.

(2) As you can see, the Ts and Os in equation 14 are unitless. In the paper, in order to evaluate the effect of model training, the training error expression in equation 14 is used: $E = \frac{1}{S}\sum_{s}^{S}(T_s - O_s)^2$, where S is the number of training samples, and Ts and Os are the expected output value and predicted output value of training sample S, respectively, which as the theoretical value f(x) and the predicted value f(x)` of the *Rosenbrock and Bohachevsky* testing functions in the Testing Algorithm part , therefore Ts and Os only represent values without units.

(1) *Rosenbrock* function:

$$f_1(x) = 100 \times (x_1^2 - x_2)^2 + (1 - x_1)^2, x_i \in [-10, 10], i = 1, 2 \tag{1}$$

(2) *Bohachevsky* function:

$$f_2(x) = x_1^2 + x_2^3 - x_1 x_2 x_3 + x_3 - \sin(x_2^2) - \cos(x_1 x_3^2), x_i \in [-2\pi, 2\pi], i = 1, 2, 3 \tag{2}$$

3- Considering the y-label of Fig. 4, I can not see any explanations related to the Rastrigin benchmark function used in your theoretical case. It would be nice if you also provided a clear version of the manuscript.

(3) We are very sincerely sorry that the Rosenbrock and Bohachevsky testing functions have not been explained in detail in the theoretical case. In order to facilitate the reader's understanding of the article, we have made corresponding modifications in the Algorithm Testing section as follows:

In order to investigate the COPSO-BP performance and reliability, Rosenbrock and Bohachevsky testing functions were adopted, which are typical non-convex functions and mainly to evaluate the performance of unconstrained algorithms. However, due to the random nature of the function, it is not easy to solve and has a global minimum function value f(x) =0.

Moreover, the Rosenbrock function is described in more detail: Rosenbrock function is a very classic function test problem in unconstrained optimization theory and method, it is an important tool to measure the advantages and disadvantages of unconstrained algorithms. In the field of numerical optimization, the function was proposed by Howard.H. Rosenbrock in 1960. In addition, it is a typical non-convex function that mainly used to optimize the performance test of the algorithm, but it is not easy to solve. And the fitness of the function is very simple in the terrain away from the most advantageous area, but the area near the most advantageous is banana-shaped. There is a strong correlation between variables, and gradient information often misleads the search direction of the algorithm. Each contour of the Rosenbrock function is roughly parabolic, and its global minimum is also in a parabolic valley (banana-type valley). It is easy to find this valley, but which is quite difficult to find the minimum of the whole domain that since the value in the valley does not change much. Among them, the Rosenbrock function could find the global minimum value of 0 at $(x_1, x_2,..., x_n) = (1,1,...,1)$, but due to its random nature, any optimization algorithm based on the falling gradient fails to find the global minimum value. A three-dimensional map of the Rosenbrock function of two of these variables is shown in Figure 1.

[Figure]

Fig.1 3D diagram of the Rosenbrock function

4- You declare that Fig. 3 shows one of the evolutionary training error curves. Considering the sentence given below, you should clearly verify whether this finding is also valid for each independent run of the algorithms.

"However, the COPSO-BP has better convergence rate and optimization efficiency in the early stage in Fig.3."

(4) In the Algorithm testing part of the paper, in order to verify the accuracy and stability of the algorithm, 20 independent experiments were carried out. The experiments show that the results predicted by the COPSO-BP algorithm are better than those of SPSO-BP in almost all experiments. Figure 2 below shows the results of the three runs of the Rosenbrock function by the SPSO-BP and the COPSO-BP. And from Figure 2 and Table 1, it can be seen that after 20 independent experiments, whether it is the Rosenbrock test function or the Bohachevsky test function, the optimal value and average value predicted by the COPSO-BP algorithm are better than SPSO-BP results, so that COPSO-BP has better convergence rate and optimization efficiency. However, the length of the article is limited, and it is not listed one by one. The corresponding supplements in the manuscript are as follows:

One of the evolutionary training error curves (select one in 20 times randomly) were shown in Fig.3, .........

[Figure]

**Fig. 2** Training error curves of SPSO-BP and COPSO-BP algorithms

**Table.1** Comparison of SPSO-BP and COPSO-BP algorithm for testing functions

| Testing functions | SPSO-BP | | COPSO-BP | |
|---|---|---|---|---|
| | Average value | Optimal value | Average value | Optimal value |
| *Rosenbrock* | 2.375e-3 | 2.300e-5 | 1.201e-3 | 2.410e-06 |
| *Bohachevsky* | 0.225 | 1.024e-3 | 0.193 | 3.360e-4 |

5-Please update Fig. 12 by adding the noise content to the theoretical curve.

(5) For the rigor of the paper, according to your suggestion, Fig.3 in the manuscript has been updated as follows:

[Figure]

**Fig.3** Forward data of Hz and data with 5% noise

6- How did you obtain error values less than the noise content added to the theoretical data?

(6) The relative error values in the paper are defined as follows, where the total relative error is the sum of the relative errors which of the resistivity and layer thickness parameters of the layered geoelectric model inversion, and the following is added to the '**3-layered H type model**' section of the manuscript:

The relative error is defined as

$$Err_{\_rel} = \left| \frac{T^*_{\_cal} - O^*_{\_ref}}{O^*_{\_ref}} \right| \qquad (3)$$

where $T^*_{\_cal}$, $O^*_{\_ref}$ are the calculated and reference value for the geoelectric models.

Furthermore, in order to verify the robust performance of the proposed algorithm, 5% and 10% noise are added to the theoretical data, and then inverted by the BP and COPSO-BP. Since the trained BP neural network can correctly predict for the trained mode or noisy mode, that is, BP neural network has the generalization ability to apply learning results to new knowledge. In addition, BP neural network does not have a great impact for global training results after its partial or partial neurons are destroyed, that is, BP neural network has a certain fault tolerance. Therefore, even for theoretical data with large noise, it can get better results by inversion, and then get the total relative error. Using the chaotic oscillation inertia particle swarm optimization algorithm (COPSO) with accelerated convergence to correct the initial parameters of BP that can avoid its local optimization, so that strengthening its generalization ability, fault tolerance and anti-noise performance. The relative error obtained inverting the noise-containing data by the COPSO-BP algorithm is smaller than these of the BP algorithm inversion, as shown in Table 2.

**Table 2** Comparison of inversion results for three-layer H type (with noise) model

| model parameters | | resistivity $\rho(\Omega \cdot m)$ | | | thickness h(m) | | Total relative error(%) |
|---|---|---|---|---|---|---|---|
| | | $\rho_1$ | $\rho_2$ | $\rho_3$ | $h_1$ | $h_2$ | |
| true value | | 100 | 10 | 100 | 100 | 200 | -- |
| without noise | BP | 99.724 | 9.937 | 100.765 | 99.031 | 198.701 | 3.284 |
| | COPSO-BP | 100.031 | 9.991 | 99.310 | 100.234 | 200.886 | 1.487 |
| 5% noise | BP | 101.374 | 9.966 | 98.283 | 101.255 | 199.282 | 5.039 |
| | COPSO-BP | 100.252 | 9.977 | 98.222 | 101.206 | 199.228 | 3.847 |
| 10% noise | BP | 90.525 | 9.931 | 99.481 | 101.748 | 203.105 | 13.976 |
| | COPSO-BP | 104.472 | 9.96050 | 101.345 | 100.570 | 199.437 | 7.064 |

7-Considering Fig. 14 that present the results of two approaches, the same resistivity range and contour interval must be used.

(7) Special thanks to your careful review. For the rigor of the article, we modified Fig.14 in the manuscript to use the same resistivity range and contour spacing as shown below.

[Figure]

(a) BP            (b) COPSO-BP

**Fig. 4**. Inversion results of BP (a) and COPSO-BP (b).

8-You should also mention in the paper which model parameters did you obtain through the PSO part of the algorithm at the end of 30 iterations. What are their error values as initial estimates requiring by the BP approach?

(8) In the research of this paper, because the BP neural network is very sensitive to the initial network weight, the network is initialized with different weights, which tends to converge to different local minimums, resulting in different results for each training. In view of the fact that the neural network is sensitive to the initial weight and easy to fall into the local minimum, the heuristic global search particle swarm optimization algorithm (PSO) with simple structure, fast convergence and high precision is used to optimize the initial weight and threshold of the neural network. That is, after 30 iterations of the PSO algorithm, the initial weights and thresholds of the BP neural network are obtained, and then the COPSO-BP is trained, so that the inversion parameters of the geoelectric model are not easy to fall into the local optimum, resulting in better model parameter results. In order to facilitate the reader's understanding, the '**BP Neural Network with COPSO algorithm**' section in the manuscript supplements the following:

therefore, the COPSO algorithm is proposed to optimize the initial weight and threshold of BP.

For different research models, the initial estimation error of the BP method (the following formula 4) is usually different, and the BP neural network itself is a black box. It is difficult to confirm the initial estimation value, which is random, so the paper does not give a clear initial estimated error value. However, according to a large number of simulation experiments, the initial estimation mean square error(MSE) of the model parameters for the layered geoelectric model studied in this paper is within 1, as shown in Fig. 5.

The formula for calculating the $i$-th particle fitness is defined as

$$f_i = \frac{1}{S}\sum_{s=1}^{S}\sum_{j=1}^{m}\left(Y_{sj} - \hat{Y}_{sj}\right)^2 \tag{4}$$

where $S$ is the number of training set samples, $m$ is the output neurons number, $Y_{sj}$ is the $j$-th true output of the $s$-th sample, and $\hat{Y}_{sj}$ is the corresponding predict output.

[Figure]

**Fig. 5** Mean square error curves comparison for three-layer H type and five-layer KHK type geoelectric model

9-The title of the paper should be reconsidered by the authors. What does it mean "Transient Electromagnetic Inversion". Think about "A fast approximation for 1D Inversion of Transient Electromagnetic Data by .." or etc.

(9) Sincerely thank you for your comments. Compared with the title of the article in the previous manuscript, we agree that your comments on the title of the article are very pertinent and more suitable for the content of the study, so we modify the following based on your comments:

A fast approximation for 1D Inversion of Transient Electromagnetic Data by BP Neural Network and improved Particle Swarm Optimization.

Special thanks to you for your good comments.

Reviewer #2:

Dear reviewers:

Comments:

For final publication, the manuscript should be accepted as is.

Special thanks to you for your good support.

We tried our best to improve the manuscript and made some changes in the manuscript. These changes will not influence the content and framework of the paper. And here we did not list the changes but marked in red in revised paper.

We appreciate for Editors/Reviewers' warm work earnestly, and hope that the correction will meet with approval.

Once again, thank you very much for your comments and suggestions.

[revised manuscript text omitted]